# See What LLMs Cannot Answer: A Self-Challenge Framework for Uncovering LLM Weaknesses

**Yulong Chen**[1,2*]  **Yang Liu**[3]  **Jianhao Yan**[1]  **Xuefeng Bai**[1]  **Ming Zhong**[4]
**Yinghao Yang**[1]  **Ziyi Yang**[3]  **Chenguang Zhu**  **Yue Zhang**[1,5†]
[1] Westlake University   [2] University of Cambridge   [3] Microsoft GenAI
[4] UIUC   [5] Westlake Institute for Advanced Study
yulongchen1010@gmail.com  yaliu10@microsoft.com  yue.zhang@wias.org.cn

## Abstract

The impressive performance of Large Language Models (LLMs) has consistently surpassed numerous human-designed benchmarks, presenting new challenges in assessing the shortcomings of LLMs. Designing tasks and finding LLMs' limitations are becoming increasingly important. In this paper, we investigate the question of whether an LLM can discover its own limitations from the errors it makes. To this end, we propose a Self-Challenge evaluation framework with human-in-the-loop. Starting from seed instances that GPT-4 fails to answer, we prompt GPT-4 to summarize error patterns that can be used to generate new instances and incorporate human feedback on them to refine these patterns for generating more challenging data, iteratively. We end up with 8 diverse patterns, such as text manipulation and questions with assumptions. We then build a benchmark, SC-G4, consisting of 1,835 instances generated by GPT-4 using these patterns, with human-annotated gold responses. The SC-G4 serves as a challenging benchmark that allows for a detailed assessment of LLMs' abilities. Our results show that only 44.96% of instances in SC-G4 can be answered correctly by GPT-4. Interestingly, our pilot study indicates that these error patterns also challenge other LLMs, such as Claude-3 and Llama-3, and cannot be fully resolved through fine-tuning. Our work takes the first step to demonstrate that LLMs can autonomously identify their inherent flaws and provide insights for future dynamic and automatic evaluation.

## 1 Introduction

Large Language Models (LLMs), such as GPT-4 (OpenAI, 2023) and Llama (Touvron et al., 2023a; Dubey et al., 2024), have shown remarkable performance on diverse Natural Language Processing (NLP) tasks, and have been trusted by users as search engines and personal assistant due to its great capacity (Tan et al., 2023b). To better understand the capability of LLMs, much effort has been dedicated to evaluating LLMs on multiple benchmarks (Fabbri et al., 2021; Gao & Wan, 2022; Huang et al., 2023; Bang et al., 2023; Zhong et al., 2023). Early LLM evaluation work follows traditional evaluation protocols, which are designed to evaluate task-specific models on datasets of single tasks (Nallapati et al., 2016; Rajpurkar et al., 2016; Goyal et al., 2022), or benchmarks that assemble multiple tasks for evaluating a certain capability, e.g., math problems (Cobbe et al., 2021) or summarization (Chen et al., 2023). More recent work evaluates LLMs on complex and expertise-level human tasks (Zhong et al., 2023; Skopek et al., 2023; Wang et al., 2024; Gandhi et al., 2024), such as the lawyer qualification test (Shui et al., 2023), GAOKAO test (Zhang et al., 2023b), etc.[1]

---

*Yulong Chen completed this work during his internship at Microsoft when he was a D.Phil. student at Westlake University.

†Yue Zhang is the corresponding author.

[1]Due to space limit, we present more Related Work in Appendix A.

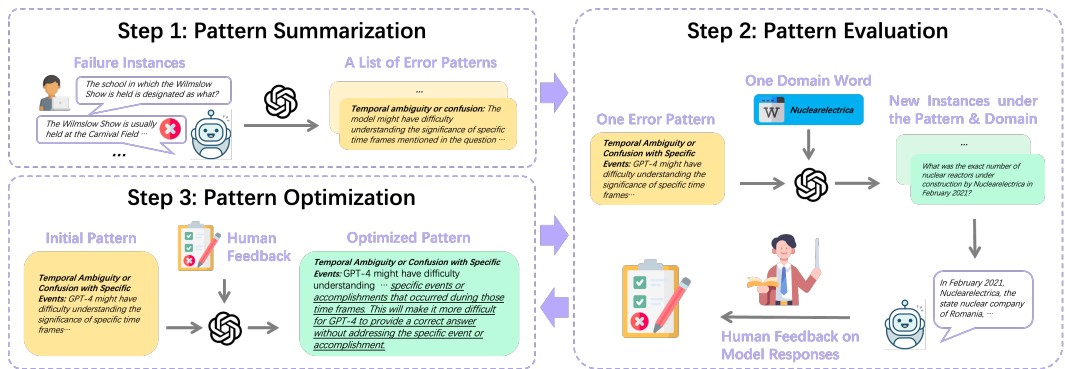

Figure 1: The overall Self-Challenge framework. We first summarize initial error patterns from seed failure instances (Step 1). Then, we perform pattern evaluation (Step 2) to determine whether summarized patterns can be used to generate challenging queries, and obtain corresponding human feedback; pattern optimization (Step 3) to modify the original pattern, making it more accurately describe challenging features, based on human feedback (the difference between the initial pattern and optimized pattern in is highlighted by *underlined* text). We frame Step 2 and Step 3 iteratively.

Although benchmarking scores are widely used to assess the capabilities of LLMs, they offer a somewhat *superficial* view, i.e., mostly indicating whether LLMs can perform a specific task without deeper analyses into the systems to uncover the precise reasons for failure or the nuanced aspects of performance degradation (Zhang et al., 2023b). As a result, people can be satisfied with high evaluation scores, but overlook the model's real weaknesses reflected in the negative instances, which can pertain to more fundamental issues, such as tokenization (Sennrich et al., 2015). Therefore, developing more dynamic, adaptive, and automated evaluation methods that can efficiently detect the limitations of LLM capabilities remains an open challenge for the research community.

To mitigate the above, we propose a Self-Challenge[2] framework that allows dynamic and interpretable evaluation for LLMs. The key idea is to prompt LLMs to identify their own limitations based on the errors they make, and then generate queries featuring those limitations to challenge themselves. As shown in Figure 1, given a set of failure instances from an LLM, we first prompt the LLM to summarize error patterns (Step 1), and then ask it to iteratively evaluate (Step 2) and refine these patterns (Step 3), discovering new challenging features and detailing those features. During this process, we engage human experts to evaluate the quality of summarized patterns by manually identifying whether such patterns can be used to generate challenging queries. Such a framework offers two advantages: First, it enables the dynamic generation of large-scale datasets for quantitative evaluation, starting from a small set of qualitative instances. Thus, it is adaptive and free of human design for tasks. Second, the patterns provide a detailed description of features that can lead to the failure of LLMs, which allows more fine-grained evaluation and thereby makes the weaknesses of LLMs more interpretable.

Employing the Self-Challenge framework on GPT-4 (OpenAI, 2023), we obtain 8 challenging patterns from 189 failure instances. Those error patterns are distinct, covering different aspects of GPT-4' limitations, such as *Text Manipulation or Transformation* and *Temporal Ambiguity or Confusion with Specific Events*. Leveraging these patterns, we construct a new benchmark, SC-G4, consisting of 1,835 challenging instances in the combination of 8 patterns and 30 domains, where GPT-4 can only answer 44.96% correctly.

We further investigate characteristics of those error patterns using SC-G4. We first benchmark multiple LLMs in a zero-shot setting and show that those error patterns from GPT-4

---

[2]Later in this paper, unless otherwise specified, the terms "*challenge*" or "*challenging*" refer to behaviours or features of queries that can induce LLMs to generate incorrect, hallucinatory, or non-factual responses.

can generalize across different LLMs such as Claude–3 only achieving 24.47% accuracy and Llama-3-70B achieving 23.32% accuracy. Interestingly, our pilot fine-tuning experiments show these errors cannot be reliably addressed by simply fine-tuning on SC-G4, suggesting that part of them can be "bugs" of LLMs, such as incompetence of text manipulation on sub-word and character levels.

We release our data at https://github.com/cylnlp/Self-Challenge-GPT4.

## 2 Self-Challenge Framework

In this section, we introduce the Self-Challenge framework, designed to identify and summarize error patterns in LLMs from errors. Additionally, it enables the generation of representative data under each pattern, facilitating quantitative evaluation for various LLMs

The overall process of Self-Challenge is shown in Figure 1. In particular, given a set of instances, the LLM is first prompted to discover potential challenging patterns from the instances (subsection 2.1). To ensure the generated patterns are of high quality, we perform pattern evaluation (subsection 2.2) and pattern optimization (subsection 2.3) with human-in-the-loop. We further frame the above two processes in an iterative manner (subsection 2.4). Appendix B presents detailed prompt information.

### 2.1 Pattern Summarization

As shown in Step 1 in Figure 1, given a set of queries $Q = \{q_1, ..., q_{|Q|}\}$, we obtain LLM responses $R = \{r_1, ..., r_{|R|}\}$ ($|Q| = |R|$) by directly prompting the LLM with the query, i.e.:

$$r_i = \phi(q_i), \tag{1}$$

where $q_i$ is the $i$-th query, $\phi$ is the LLM function, and $r_i$ is the LLM-generated response to $q_i$. We also collect correct responses $C = \{c_1, ..., c_{|C|}\}$, where $c_i$ is the correct response to $q_i$.

Then, we prompt the LLM to summarize potential error patterns from those instances:

$$P = \phi(Prompt_{summ}(Q, R, C)), \tag{2}$$

where $Prompt_{summ}$ is a natural language instruction that guides the LLM to generate initial patterns and $P$ is the summarized pattern list ($P = \{p_1, ..., p_{|P|}\}$).

### 2.2 Pattern Evaluation

In our experiments, we notice that some initially summarized patterns can be coarse-grained and of low quality, i.e., a generated pattern fails to capture the features of queries that challenge LLMs. Therefore, it is essential to evaluate these patterns. However, since there are no fixed answers or gold standards for what constitutes a good pattern, evaluating them *directly* (e.g., by comparing the similarity of two pieces of text) is not feasible.

To address this, we perform the pattern evaluation *indirectly*. In particular, we evaluate whether a pattern accurately summarizes and describes the challenging features of a set of queries by determining if it can be used to generate new, analogous, and challenging queries. Therefore, given a summarized pattern $p$, we prompt the LLM to generate new queries in a specific domain $d$,[3] i.e.:

$$Q^{p,d} = \phi(Prompt_{gen}(p, d)), \tag{3}$$

where $Q^{p,d}$ is the new query set under the pattern of $p$ in the domain of $d$, and $Prompt_{gen}$ is a natural language instruction.

Similarly, for each query in $Q^{p,d}$, we obtain its model-generated response by directly prompting the LLM (Equation 1). Human annotators then provide feedback. Each human feedback

---

[3]We find that, without the domain information (i.e., $Q^p = \phi(Prompt_{gen}(p))$, the generated queries tend to converge and lack diversity.

is a natural language that describes whether the query successfully challenges the LLM, which functions both as an evaluation score and a "*gradient*" signal for later pattern optimization. Finally, we can evaluate the pattern quality by accuracy, i.e., the portion of how many queries successfully challenge the LLM.

The detailed annotation information can be found in Appendix C.

### 2.3  Pattern Optimization

After evaluating the pattern and obtaining the feedback, we then prompt the LLM to optimize it, making it more accurate and detailed.

Taking each pattern $p$ and its newly generated queries $Q^p$ ($Q^p = \bigcup_{d \in D} Q^{q,d}$, where $D = \{d\}$), corresponding LLM-generated responses $R^p$ and human feedback $F^p$ as input, we prompt the LLM to optimize the pattern so that the optimized pattern can better reflect the features of challenging queries and be used for generating new queries:

$$p' = \phi(Prompt_{opt}(p, Q^p, R^p, F^p)), \tag{4}$$

where $p'$ is the optimized pattern and $Prompt_{opt}$ is a natural language instruction.

### 2.4  Iterative Self-Challenge

We iterate the pattern evaluation and optimization processes to ensure that the optimized patterns are of high quality. In particular, we replace $p$ with $p'$ in Equation 3 and can obtain a new set of queries generated by $p'$. Then we evaluate the query-response under $p'$ and obtain corresponding human feedback to optimize $p'$ by replacing $p$ with $p'$ in Equation 4. Through this iterative process, we can continually optimize the patterns until they are considered good enough.

Finally, we can obtain a set of high-quality patterns, which illustrate where and how LLMs can fail and can be used to generate more diverse queries under corresponding patterns.

## 3  Discovering Error Patterns in GPT-4 and Constructing SC-G4 Benchmark

To demonstrate the usage and facilitate future research, we apply our *Self-Challenge* framework to *GPT-4* (OpenAI, 2023).[4] We first discover 8 error patterns from 189 failure instances and construct a new benchmark, *SC-G4*, for quantitative evaluation.

### 3.1  Seed Instance Collection

We collect 189 diverse queries from online[5] and from previous studies that qualitatively evaluate GPT-4 such as (Zheng et al., 2023b). These instances are diverse in terms of task formats and domains, which cover traditional NLP tasks such as multi-hop QA (Zheng et al., 2023b), and non-traditional NLP tasks such as word sorting (bench authors, 2023). Appendix D shows the collection details.

### 3.2  Error Pattern Discovery

Given the initial collection of queries, we first prompt GPT-4 to discover their initial patterns. Due to the length limit, we group the instances according to their sources and prompt GPT-4 individually. In this way, we obtain a list of 14 initial patterns.

We then perform the pattern evaluation and optimization for two rounds. To smooth model bias towards a certain domain, we provide 4 domain words. For each combination of pattern

---

[4]https://platform.openai.com/docs/models/gpt-4-and-gpt-4-turbo, gpt-4-32k version.
[5]For example: https://twitter.com/home and https://github.com/manyoso/haltt4llm.

| ID | Pattern Name | # | % |
|---|---|---|---|
| A | Assumption of Existence | 179 | 9.75 |
| B | Bias towards More popular or Well-known Information&Overgeneralization | 221 | 12.04 |
| C | Complex Counting or Identification Tasks | 232 | 12.64 |
| D | Complex Logical Reasoning and Paradoxes | 229 | 12.48 |
| E | Complex Syntactic Structures & Multiple Clauses | 256 | 13.95 |
| F | Recursive and Unusual Patterns | 230 | 12.53 |
| G | Temporal Ambiguity or Confusion with Specific Events | 217 | 11.83 |
| H | Text Manipulation or Transformation | 271 | 14.77 |

Table 1: A list of names of patterns, with instance counts (#) and portions (%) for each individual pattern.

and domain, we generate 10 new queries, and thus have $4 \times 10 = 40$ new queries for each pattern. Then, we manually evaluate model responses with human feedback. During this process, we discard patterns that are consistently found less satisfying, i.e., over 50% of generated queries cannot challenge GPT-4. Finally, we obtain 8 high-quality patterns.

Table 1 presents the list of pattern names, where we see that the error patterns cover different aspects that can challenge GPT-4, ranging from underlying problems such as character (letter) processing, which can be related to tokenization issues, to more high-level problems such as logical reasoning.

Figure 2 presents one typical error pattern and its corresponding original pattern, where we highlight their *difference*. We see that the original patterns are general while the optimized patterns are more comprehensive and detailed in describing what kind of specific features should be included in the query, such as "*such as dependency or constituency...*". Moreover, the optimized patterns include instructions on how to re-produce queries under such patterns.

We also apply the Self-Challenge framework to Llama-3-70B (Dubey et al., 2024) for pattern discovery as shown in Appendix G, which, however, is less effective compared with GPT-4.

### 3.3 The SC-G4 Benchmark

After obtaining the patterns, we construct the SC-G4 benchmark consisting of 1,835 queries with human-annotated gold responses under the combination of 8 patterns and 30 domains.

We randomly select 30 Wikipedia titles from Wikipedia metadata (Appendix E). For each combination of pattern and title, we prompt the LLM to generate 10 diverse queries at once. Thus, we have $8 \times 30 \times 10 = 2,400$ new queries in total. Generally, we find that new queries can mostly follow the pattern and are diverse in terms of specific tasks. Take pattern C (*Complex Counting or Identification Tasks*, Appendix F) for example, the new query: "*In the sentence, 'Joey Votto is known for his exceptional plate discipline and ability to get on base,' how many words have the letter 'e' as the third character and end with the letter 'n'?*" suits the requirement of "*identify specific elements in complex sentences*". Moreover, the task of *identifying a letter that comes after another letter* is not shown in the aforementioned instruction. It suggests we can obtain diverse queries out of the limited features described in patterns.

As we aim to assemble the benchmark by collecting self-challenging data generated by GPT-4, there remain two major questions. First, a few generated queries can still distract from corresponding patterns, in particular when we use new domains for generation. Second, the generated queries can lead to open-ended responses such as "*Please combine two names of the centers and institutes at Stanford University in an alphabet order*", which poses challenges for practical evaluation. Therefore, we further filter out 565 queries to ensure the remaining 1,835 queries can be used to evaluate LLMs by comparing model-generated responses with responses annotated by humans. We ask 4 human annotators to write the gold response with the help of GPT-4, where annotators also manually give a binary label to the GPT-4

> **Original Pattern:**
> Complex Syntactic Structures and Multiple Clauses: GPT-4 struggles to accurately parse sentences with complex structures and multiple clauses, leading to incorrect identification of specific words or relationships between words.
>
> To generate questions that challenge GPT-4, create sentences with multiple clauses, embedded phrases, and complex relationships between words and ask GPT-4 to parse the sentence or identify specific words or relationships between words. This will make it difficult for GPT-4 to parse the sentence accurately and identify the correct answer

> **Original Pattern:**
> Complex Syntactic Structures and Multiple Clauses: GPT-4 struggles to accurately parse sentences with complex structures and multiple clauses, leading to incorrect identification of specific words or relationships between words.
>
> To generate questions that challenge GPT-4, create long and complex sentences with multiple clauses, embedded phrases, and complex relationships *(such as dependency or constituency) between words. Then, ask GPT-4 to parse the sentence, identify specific words or dependency or constituency relationships between words, or rephrase the sentence.*
>
> *For example: 1. \"I'm a student very close to the VCU area that's looking for an apartment or house to move in to with another person/people.\" What are the words in the previous sentence, which form a constituency with the second word that begins with letter 'a'?*
>
> *2. \"Unfortunately, I currently have no funding or capacity to advise interns or visiting students. Please consult the EdiNLP web site for more information on potential supervisors.\" What are the words in the previous sentence, which hold a dependency with \"funding\" or what are the predicative words in the above sentences?*
>
> This will make it difficult for GPT-4 to parse the sentence accurately and identify the correct answer. *Avoid fact-based questions that do not require complex linguistic analysis.*

Figure 2: A case of optimized pattern, coupled with its original pattern. We highlight their *difference*. Full patterns can be found in Appendix F.

response, indicating whether it is correct or not. This binary label is further used as the gold evaluation results of GPT-4 on SC-G4. The detailed annotation information can be found in Appendix C.

Finally, we obtain 1,835 queries annotated with gold responses. Our human evaluation shows that GPT-4 can only achieve an accuracy of 44.96%. Table 2 gives the basic statistics for the overall SC-G4 benchmark. This suggests that most queries generated by GPT-4 can successfully challenge GPT-4 themselves. Also, we see that compared with GPT-4 responses, the averaged token lengths of human-annotated responses are much less. It shows most GPT-4 responses are further corrected by human annotators.

## 4 Benchmarking LLMs on SC-G4 and Investigating Error Generalization across LLMs

In this section, we investigate the main question of *whether GPT-4 error patterns are universal and can generalize across other LLMs*.

### 4.1 Experimental Setup

In addition to GPT-4, we experiment with other closed-source LLMs and open-source LLMs on SC-G4, in both zero-shot and few-shot settings

**Baseline Systems.** We compare Gemma-7B (Team et al., 2024), Phi3-7B-instruct (Abdin et al., 2024), Mixtral-8×7B-instruct (Jiang et al., 2024), Gemini-1-pro (Team et al., 2023), Llama models (Touvron et al., 2023b; Dubey et al., 2024), Claude-3-Sonnet,[6] and GPT-3.5-turbo and GPT-4o models (Achiam et al., 2023; Brown et al., 2020) with GPT-4 (OpenAI, 2023).

**Evaluation Data.** For zero-shot experiments, we evaluate LLMs on the whole SC-G4 data. For few-shot fine-tuning, we experiment under 2 settings as shown in Table 2. (1) **Split randomly:** We first randomly split the SC-G4 into train/dev/test sets (500/335/1000), where train and test sets share common patterns and domains. (2) **Split by domains**: We split the SC-G4 into train/dev/test sets by constraining domains in them. In other words, train and test sets have different domains but share common patterns. In particular, we split data of 10/5/15 domains into train/dev/test sets (629/247/959), respectively.

---

[6] https://www.anthropic.com/news/claude-3-family

| Self-Chal | # | Avg Query | Avg Gold R | Avg GPT-4 R |
|---|---|---|---|---|
| Overall | 1,835 | 34.84 | 49.08 | 78.13 |
| Random-set | 500/335/1,000 | 34.07/35.37/35.05 | 48.56/48.44/49.56 | 77.48/77.00/78.84 |
| Domain-set | 629/247/959 | 34.34/35.60/34.98 | 43.21/56.57/51.01 | 79.05/78.12/77.54 |

Table 2: Basic statistics for SC-G4 benchmark. **#**: instance count. Avg: averaged token length. R: Response. We also report statistics of different data split strategies (Random and Domain) as discussed in subsubsection 4.2.2.

**Implementation Details.** For zero-shot experiments, we directly prompt baseline LLMs given queries as input. For few-shot experiments, we use LoRA (Hu et al., 2021) to fine-tune the Llama 2-Chat (7B) in different settings (c.f. subsubsection 4.2.2 for detailed settings of different data split strategies). We fine-tune the model on 8 A100 GPUs on our training sets for 5 epochs[7] and choose the best checkpoint by their loss on validation sets. We use the AdamW optimizer (Loshchilov & Hutter, 2017) and set the learning rate as 2e-4.

**Evaluation Protocol.** For GPT-4, we directly evaluate their performance when annotating the gold responses (subsection 3.3). However, it is not practical to conduct manual evaluation for all models, which can be costly. Therefore, we evaluate model performance via prompting GPT-4 (Liu et al., 2023) to compare the model response and gold response, i.e.:

$$l = \phi(Prompt_{eval}(q, h, r)), \tag{5}$$

where $l$, $q$, $h$ and $r$ are output label ("*correct*" or "*incorrect*"), query, gold and model-generated responses, respectively. Appendix B presents the prompt in detail. We show that this evaluation method achieves a moderate-to-high $F$-1 score of around 90 in Appendix I.

## 4.2 Result and Analysis

We first show results in a zero-shot setting, finding the "bugs" are universal to other LLMs, and present few-shot results, finding most error patterns cannot be simply solved via tuning.

### 4.2.1 Zero-shot Performance

Table 3 shows the model performance on the full SC-G4 data in the zero-shot setting. Generally, GPT-4 achieves the best performance of 44.96% human-evaluated accuracy and 47.63 GPT-4 evaluated accuracy, which, however, is still less satisfying. Compared with GPT-4, other LLMs show poor performance on our data, where GPT-4o outperforms other LLMs including Turbo, Llama models, and Claude-3. This suggests that although those instances are initially aimed to challenge GPT-4, they are difficult for other LLMs. For Llama 2 based models, Chat models slightly outperform standard models. Also, we see that as the sizes of LLMs increase, the overall performance increases.

We further analyze the model performance under different patterns, respectively. The breakdown analysis is shown in Table 3. Note that the pattern IDs are aligned with as in Table 1 and Appendix F, respectively. First, GPT-4 achieves the best human-evaluated accuracy on patterns D (68.12%), A (67.60%) and B (65.61%), and show very poor performance on patterns H (9.23%), C (22.41%) and G (38.25%). Also, Turbo shows a similar trend on those patterns, while the performance on each pattern is much poorer than GPT-4. Interestingly, although outperforming Turbo on most patterns, GPT-4o shows poor performance on pattern A. We assume that this can be because the multi-modal training does not necessarily enhance the models' ability to disambiguate hallucinatory queries. In addition, Llama-based models and other LLMs underperform GPT-4 on all patterns.

For Llama models, it is noted that Chat models show very poor performance on pattern A compared with standard models, indicating they are more prone to hallucinate on queries that include non-existent entities. Also, we see that Llama models of bigger size do not

---

[7]We also trained models with more steps, but did not observe large performance improvement.

| LLMs | A | B | C | D | E | F | G | H | all |
|---|---|---|---|---|---|---|---|---|---|
| Gemma-7B | 25.70 | 8.14 | 8.19 | 11.35 | 16.80 | 6.09 | 5.07 | 0.00 | 9.65 |
| Gemini-1-pro | 17.88 | 17.65 | 7.76 | 17.03 | 26.17 | 10.87 | 5.07 | 0.00 | 12.59 |
| Phi3-7B-instruct | 12.29 | 21.72 | 15.09 | 37.55 | 20.31 | 21.30 | 9.22 | 2.21 | 17.33 |
| Llama-2-7B | 0.56 | 19.46 | 4.74 | 13.54 | 22.66 | 4.78 | 6.45 | 0.00 | 9.21 |
| Llama-2-7B-chat | 8.94 | 15.84 | 1.72 | 3.06 | 19.92 | 3.91 | 7.83 | 1.11 | 7.74 |
| Llama-2-13B | 0.00 | 23.53 | 6.90 | 14.41 | 27.34 | 5.65 | 11.06 | 0.00 | 11.34 |
| Llama-2-13B-chat | 17.88 | 18.55 | 5.17 | 13.97 | 22.27 | 5.65 | 7.83 | 0.37 | 11.17 |
| Llama-3-70B-instruct | 43.58 | 31.22 | 19.40 | 30.57 | 32.81 | 20.87 | 15.21 | 0.37 | 23.32 |
| Mixtral-8x7B-instruct | 24.58 | 24.43 | 8.19 | 18.34 | 26.56 | 13.04 | 7.37 | 0.37 | 14.93 |
| Claude-3-Sonnet | 41.34 | 34.39 | 20.69 | 35.37 | 29.69 | 24.35 | 15.21 | 1.85 | 24.47 |
| GPT-3.5-turbo | 49.72 | 42.53 | 14.22 | 39.74 | 43.75 | 23.91 | 23.04 | 2.58 | 28.94 |
| GPT-4o | 30.73 | 41.63 | 30.60 | 44.54 | 43.36 | 35.22 | 20.74 | 8.86 | 31.66 |
| GPT-4 | 65.92 | 69.68 | 22.84 | 71.18 | 60.16 | 46.52 | 43.32 | 11.44 | 47.63 |
| GPT-4† | 67.60 | 65.61 | 22.41 | 68.12 | 57.42 | 41.74 | 38.25 | 9.23 | 44.96 |

Table 3: Model performance on full SC-G4 benchmark. We show the overall performance and breakdown performance under different individual patterns. † indicates human evaluation while others are evaluated by GPT-4. The IDs of patterns correspond to IDs in Table 1.

| ID | # | G-4$^\dagger$ | G-4 | L-2 | L-2$^*$ | ID | # | G-4$^\dagger$ | G-4 | L-2 | L-2$^*$ |
|---|---|---|---|---|---|---|---|---|---|---|---|
| A | 106 | 65.09 | 60.38 | 7.55 | 31.13↑ | A | 98 | 67.35 | 67.35 | 7.14 | 35.71↑ |
| B | 110 | 64.55 | 69.09 | 15.45 | 10.00↓ | B | 112 | 73.21 | 76.79 | 16.96 | 11.61↓ |
| C | 129 | 20.16 | 23.26 | 1.55 | 6.20↑ | C | 124 | 20.97 | 20.16 | 2.42 | 4.84↑ |
| D | 118 | 66.95 | 71.19 | 1.69 | 0.85↓ | D | 130 | 70.00 | 71.54 | 2.31 | 2.31 |
| E | 145 | 56.55 | 58.62 | 19.31 | 23.45↑ | E | 125 | 63.20 | 65.60 | 22.40 | 20.00↓ |
| F | 124 | 40.32 | 45.16 | 3.23 | 5.65↑ | F | 118 | 42.37 | 47.46 | 3.39 | 2.54↓ |
| G | 118 | 34.75 | 40.68 | 9.32 | 8.47↓ | G | 113 | 42.48 | 46.02 | 10.62 | 9.73↓ |
| H | 150 | 8.67 | 10.00 | 1.33 | 0.00↓ | H | 139 | 7.91 | 10.79 | 0.72 | 0.00↓ |
| all | 1,000 | 43.10 | 45.80 | 7.40 | 10.40↑ | all | 959 | 47.24 | 49.24 | 8.03 | 10.01↑ |

Table 4: Model performance on randomly split test set (**left**) and performance on test set split by domains (**right**). G-4: GPT-4. L-2: Llama 2-Chat (7B). ∗: LoRA-tuned.

significantly perform better than smaller models on certain patterns, such as pattern H (Llama-2-7B: 0.00; Lllama-2-13B: 0.00; Llama-3-70B: 0.37). This indicates that simply scaling up the model size may not solve problems of such patterns. We present the breakdown analysis of different domains in Appendix J.

### 4.2.2 Few-shot Performance

We experiment with 2 settings as stated in Table 4.1. The first is to give a general overview of whether fine-tuning can bring improvement, while the latter is to present a more fine-grained analysis.

**Random Split.** Table 4 (left) presents the results, which also include zero-shot performance of GPT-4 and Llama 2-Chat (7B) on the test set for comparison. Generally, we see that the LoRA-fine-tuned model shows a 3.00% accuracy improvement. However, we see that fine-tuning only brings great improvement on pattern A (23.58%↑), but few on the rest. On the other hand, fine-tuning even slightly hurts the model performance, such as B (5.45%↓) and H (1.33%↓). The above results suggest that most challenges cannot be simply resolved by fine-tuning.

**Domain Split.** As shown in Table 4 (right), similarly, the LoRA-tuned model shows the best improvement on pattern A (28.57%↑) and shows poorer performance on B (5.35%↓), E (2.40%↓), F (0.85%↓), G (0.89%↓) and H (0.72%↓) than zero-shot model. Moreover, the performance gap between the LoRA-tuned and zero-shot models on this dataset is less than on the randomly split dataset. Such a narrower gap indicates that different domain further

enhances the difficulty of generalization ability. This suggests that the model does not learn relevant pattern knowledge from fine-tuning, except for pattern A.

### 4.3 Findings and Highlights

**The error patterns can be universal in different LLMs.** Evidenced by our zero-shot results, we find all LLMs show poor results on GPT-4 error patterns. The results also show that most LLMs tend to perform relatively well on patterns A and E, while relatively poor on patterns C and H. Similar trends can be consistent with previous research (Huh et al., 2024; Bommasani et al., 2022) that LLMs can be homogeneous as they are trained with similar techs (e.g., transformers (Vaswani et al., 2017)) on similar data (e.g., Wikipedia) (Touvron et al., 2023b; Radford et al.; Brown et al., 2020).

**LLMs tend to fail on tasks that are related to tokenization.** By analyzing patterns C and H, we find they are mostly related to the tokenization mechanism in LLMs. Challenging queries under these two patterns include identifying certain (sub-)words in a word, or counting words with a specific feature (e.g., "*start with 'e', and end with 'f'*"). Such tasks can be closely related to applications such as poetry generation (Oliveira, 2017), spelling and grammar checking (Soni & Thakur, 2018), and teaching aide (Felten et al., 2023). We emphasize that such issues reveal the problems of underlying tokenization mechanisms, which can further influence the performance of downstream tasks. Also, it reveals that the current tokenization design (Sennrich et al., 2015) in LLMs is different from humans, making LLMs less sensitive to character-level information. While humans can easily solve these tasks, LLMs fail to do so.

**LLMs can sense the uncertainty within themselves.** Our few-shot (in particular the cross-domain) experiments show that the Llama model can benefit most from fine-tuning on pattern A. It shows LLMs may have learned that there can be non-factual information in the query even although we do not fine-tune them on more in-domain data. We hypothesize that this reflects the LLMs' ability to sense internal uncertainty (Zhang et al., 2024; Ye et al., 2024), and that fine-tuning on such data enhances their threshold for taking uncertain queries into account and responding more cautiously. We show more analyses, including case study, in Appendix L.

**Simply fine-tuning may not be able to solve all problems.** We observe that fine-tuning can enhance model performance on certain tasks (pattern A) (Yin et al., 2023). However, as evidenced by our few-shot experiments in different settings, for the other patterns, we find that fine-tuning does not bring consistent improvement. Although we acknowledge that little improvement can be attributed to the small training data, there can be some inherent flaws in LLMs. More advanced technologies or a large amount of relevant data may help to resolve those error patterns, which we leave for future work.

## 5 Conclusion

In this work, we introduced a Self-Challenge framework that evaluates LLMs, revealing their nuanced limitations by prompting LLMs to challenge themselves iteratively. By integrating this framework with GPT-4, we obtained 8 typical error patterns, and assembled the SC-G4 benchmark with 1,835 challenging instances, where GPT-4 achieves 44.96% accuracy. We further showed that such patterns are universal across multiple LLMs, highlighted by low accuracy rates in both zero-shot and few-shot settings, suggesting that the limitations may not be mere oversights but inherent "bugs" within current LLM architectures or their pre-training data.

### Acknowledgments

We appreciate reviewers and meta-reviewers from COLM 2024 for their valuable feedback. We also appreciate Bonnie Webber, Andreas Vlachos, Leyang Cui and Rami Aly for their proofreading and insightful discussion. We acknowledge funding support from the NSFC key project 62336006.

# 6 Ethics Statement

The construction of SC-G4 involves the participation of GPT-4 (input query generation) and human annotation (gold output response).

**Data Usage and Safety:** During our annotation, we have ensured that the GPT-4 generated text is safe and does not contain any uncomfortable descriptions, such as violent crimes and events. The data in SC-G4, in particular under pattern A (*Assumption of Existence*), includes fake questions that contain fictional entities or events. This inclusion is crucial for testing and enhancing GPT-4's ability to reason about hypothetical or speculative information, thereby improving its performance across a variety of real-world applications.

**Potential Biases of GPT-4 Generated Queries:** We acknowledge that relying on GPT-4 to generate instances for evaluation can lead to potential biases, in particular data homogeneity (Ding et al., 2024; Dunlap et al., 2023). Although we enrich the input pattern with additional domain information and evaluate their level of diversity by calculating the Distinct score (Li et al., 2015) as in Appendix M, there remains a potential problem that queries under the same pattern and domain can be similar to each other without human verification. This problem can be solved by decomposing the current patterns into more fine-grained and detailed sub-patterns or providing larger domain text as a source for query generation, which we leave for future work.

**Human Annotation:** Please refer to Appendix C, where we provide a detailed annotation plan and how we compensate the annotators.

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

## A Related Work

**Evaluating LLMs.** Recently, model evaluation has gained increasing attention from the communities (Zhang et al., 2023c; Fabbri et al., 2021; Gao & Wan, 2022; Bai et al., 2023; Zhong et al., 2022; 2023; Huang et al., 2023; Bang et al., 2023; Chen et al., 2023). Before the surge of LLMs, evaluation protocols mainly focus on evaluating a certain aspect of a single task (Zheng et al., 2023b; Goyal et al., 2022). Evaluation in LLMs can be broadly divided into two categories. A line of work aims to evaluate the comprehensive performance, either on a large set of tasks (Qin et al., 2023; Bang et al., 2023; Zhong et al., 2023; Hendrycks et al., 2021a; Srivastava et al., 2023), or evaluate on representative datasets of core scenarios, such as general instruction-response (Dubois et al., 2023; Zheng et al., 2023a). Another line of research focuses on specific capabilities of LLMs, such as MATH (Cobbe et al., 2021; Hendrycks et al., 2021b), reasoning (Zellers et al., 2019; Bisk et al., 2020), instruction following (Zhou et al., 2023; Yan et al., 2024), coding (Chen et al., 2021), etc. Our work does not fall into these two categories but rather focuses on locating the specific limitations of LLMs.

There has been work that highlights the intrinsic limitation of LLMs (Yin et al., 2023; Wolf et al., 2023). For example, Yin et al. (2023) explore LLMs' self-knowledge, revealing the limits of their knowledge compared to human proficiency. Wolf et al. (2023) investigate the fundamental limitations of aligning LLM behaviours with human intentions. However,

they still focus on evaluating LLMs on certain tasks. In contrast, we find challenging tasks (error patterns) from data in the wild.

**Data Augmentation using LLMs.** Recently, researchers have explored using LLMs as annotators for creating and augmenting data of their interest, using both zero-shot and few-shot learning (Ding et al., 2024; Tan et al., 2024). For example, Dai et al. (2023); Anaby-Tavor et al. (2020); Sahu et al. (2022); Lin et al. (2023); Törnberg (2023); Zhu et al. (2023) use LLMs such as ChatGPT and GPT-4 to label raw text for text classification tasks, and Chintagunta et al. (2021); Zhang et al. (2023a) use LLMs to generate output texts for text generation tasks, such as machine translation and text summarization. Additionally, Anaby-Tavor et al. (2020) finetune GPT-2 with small data to annotate new training data for text classification tasks. Li et al. (2022); Zheng et al. (2022); Meng et al. (2023) use LLMs to augment dialogues with few-shot learning.

Our work falls in the above studies that use LLMs for data augmentation. However, different from those that mostly use LLMs to augment analogous data or label output text, we use LLMs to obtain diverse queries for a specific task under the guidance of patterns (instruction) generated by LLMs themselves.

**Using LLMs as Optimizers.** Our Self-Challenge framework involves using LLMs as optimizers to modify the patterns. Similarly, Yang et al. (2023) introduce the optimization by prompting, which employs LLMs for generating solutions in natural language. Cummins et al. (2023) explore the application of LLMs in optimizing code. Guo et al. (2023) integrate evolutionary algorithms with LLMs for discrete prompt optimization.

# B  Details of Prompts

## B.1  Summarization Prompt

We first ask GTP-4 to analyze the error instances:

> *Given the below question, incorrect model answer, and correct answer, please analyze why and how this question induces GPT-4 to generate an incorrect answer. Do not provide suggestions on how to improve GPT-4 performance on such questions.*
>
> *...*
>
> *{Question: $\{q_i\}$; GPT-4 Answer: $\{r_i\}$; Correct Answer: $\{c_i\}$}*
>
> *...*

where $\{q_i\}$, $\{r_i\}$ and $\{c_i\}$ are the *i*-th query, model response and correct answer.

Then, we ask GPT-4 to summarize possible patterns of challenging questions using the below prompt:

> *Below are some questions, where GPT-4 fails to generate correct answers, and corresponding analysis why and how the question challenges GPT-4:*
>
> *...*
>
> *{Question: $\{q_i\}$; Analysis: $\{a_i\}$}*
>
> *...*
>
> *According to the questions and analysis, please summarize the most common patterns of the above questions, which can induce GPT-4 to generate incorrect answers. Please note that:*
>
> *a. The patterns can describe features from different cognitive levels in those questions, for example, syntax, semantics, pragmatics, or even the knowledge acquisition in GPT-4.*
>
> *b. Each above question can contain multiple patterns.*
>
> *c. Each summarized pattern should at least appear in one of the above questions.*
>
> *d. For each summarized pattern, give clear and detailed explanation and analysis to make the summarized pattern understandable.*
>
> *e. Each pattern should be formulated in a reproducible way. Following each pattern, please give suggestions on: when we want to generate questions using the*

*summarized patterns, how can we generate those questions, which contain such patterns and induce GPT-4 to fail?*

where $\{a_i\}$ is the analysis to the $i$-th query.

## B.2 Generation Prompt

We use the below prompt to generate new queries:

*Below is a pattern, which is summarized from a set of questions that induce GPT-4 to generate incorrect answers.*

*Pattern: $\{P_i\}$*

*Please generate 10 new questions that contain such a pattern in the $\{domain\}$ domain. Please note that:*

*1. The question should induce GPT-4 to generate incorrect answers.*

*2. The question should be answered by a short sentence or phrase. The answer should not be complicated, for example, not a paragraph of hundreds of words.*

*3. The new questions should be diverse, and are different from each other in terms of their main contents, entities, events, and question types. Do not generate questions consistently asking the same thing, or having large lexical overlaps.*

*4. You should generate more diverse and novel questions of different types, as long as those new questions contain the pattern and successfully induce GPT-4 to generate incorrect answers.*

*5. Please generate questions with entries (e.g., "1.") and do not generate possible answers or generate patterns again.*

where $\{P_i\}$ is an individual pattern (full pattern with name and description), and $\{domain\}$ is a domain word, such as *Martin Nolan*.

## B.3 Optimization Prompt

After collecting model responses and their corresponding human feedback, we use the below prompt to optimize the initial pattern:

*Please first read the below original pattern, which is summarized from questions where GPT-4 fails, and their paired example questions.*

*Pattern: $P_i$*

*Below are new questions, which are generated using the above pattern and should induce GPT-4 to fail, GPT-4 answers to those new questions and their corresponding human feedback:*

*...*

*{Question: $\{q_i\}$; GPT-4 Answer: $\{r_i\}$; Human Feedback: $\{f_i\}$}*

*...*

*Given the new questions which are generated using that pattern, GPT-4 answers, and human feedback, please:*

*1. Check whether the new questions successfully induce GPT-4 to generate incorrect answers: if not, analyze reasons why GPT-4 can successfully generate correct answers for those new questions, which are supposed to induce GPT-4 to generate incorrect answers.*

*2. Then consider how to modify the pattern so that it can be used to generate questions where GPT-4 is more likely to fail. In particular, you should treat the questions where GPT-4 answers correctly as negative samples.*

*The modification can include but are not limited to:*

*Modify a vague pattern into a more fine-grained and specific pattern.*

*Add descriptions on what types of questions that GPT-4 can answer correctly, and how we should avoid them when generating questions.*

> *However, please ensure that questions generated by new patterns should be answerable and can be answered by a short phrase.*

where $\{q_i\}$, $\{r_i\}$ and $\{f_i\}$ are the $i$-th query, model response and human feedback.

### B.4 Evaluation Prompt

Given a query and its correct response, we evaluate the model outputs by using the below prompt:

> *Please first read the below question, its correct answer and a model-generated answer:*
> *Question: $\{question\}$*
> *Correct Answer: $\{correct response\}$*
> *Model-generated Answer: $\{model response\}$*
> *Is the model-generated answer correct or incorrect? Please only output "correct" or "incorrect".*

## C  Details of Annotation

### C.1  GPT-4 Response Verification and Gold Response Annotation

We ask annotators to write a gold response to a query with the help of GPT-4 response. In particular, given a query, we first prompt GPT-4 to generate a response and then ask the human annotator to check whether the model response is correct or not. In addition to questions and responses, annotators are also presented with corresponding patterns, to help them better understand what the task the LLMs are doing. For example, for the *Assumption of Existence* questions, responses including "*Sorry, I cannot find relevant information on [non-existent entities/events]*" would be regarded as "*correct*".

For responses that contain reasoning steps (such as text manipulation and counting tasks), they should examine whether the reasoning is correct. For example:

> Query: *In the sentence "Henrik Lundqvist is often regarded as one of the best goaltenders in the history of the NHL," identify the total number of words that end with the letter "s" (ignoring case-sensitivity).*
> GPT-4 Response: *In the given sentence, there are 3 words that end with the letter "s": "Lundqvist," "as," and "goaltenders."*
> Human Label: *incorrect* (although there are 3 words ending with "*s*", GPt-4 mistakenly take "*Lundqvist*" as a candidate)

If the response is correct, the annotator can use it as the gold response with modification. If the response is found error, the annotator is asked to perform post-editing to correct the model response, which then can be used as the gold response.

When evaluating GPT-4 responses and annotating gold responses, annotators are encouraged to utilize various tools, including search engines (e.g., Google) and syntax analyzers, to check the factuality and correctness of GPT-4 answers.

### C.2  Quality Control

To ensure quality, we require annotators to practice on 20 training samples and achieve satisfactory performance on our assessment test before proceeding to real annotations. We pay extra attention to (1) Incorrect: If any error (in particular factual) is found in GPT4 response, annotators label it as *correct* (or vice versa); annotated responses contain errors, in particular, annotators use incorrect GPT4 responses as gold responses without modification; (2) Unreadable: annotated response is oversimplified particularly if annotators modify GPT4 responses, but annotated responses need GPT4 response as context to understand.

| id | Wiki Title (domain) | # | id | Wiki Title (domain) | # |
|----|---------------------|-----|----|---------------------|-----|
| 1 | *Abney Park Cemetery* | 62 | 16 | *Martin Nolan* | 50 |
| 2 | *Anatoly Karpov* | 61 | 17 | *Maryborough Base Hospital* | 55 |
| 3 | *Chaka Khan* | 69 | 18 | *Mixtec language* | 52 |
| 4 | *Child Workers in Nepal* | 67 | 19 | *Muirchertach Ua Briain* | 56 |
| 5 | *Chinese calligraphy* | 67 | 20 | *Murgon State School* | 39 |
| 6 | *Corruption in Yemen* | 63 | 21 | *Philippine drug war* | 69 |
| 7 | *Development of COVID-19 tests* | 65 | 22 | *Prespa Agreement* | 66 |
| 8 | *Edinburgh and Northern Railway* | 72 | 23 | *Ray Cooper* | 50 |
| 9 | *Emergency medical services in Germany* | 62 | 24 | *Scouting in Illinois* | 65 |
| 10 | *Henrik Lundqvist* | 77 | 25 | *Shirley Chisholm* | 70 |
| 11 | *Joey Votto* | 67 | 26 | *Stanford University centers & institutes* | 64 |
| 12 | *John Mowbray, 3rd Duke of Norfolk* | 56 | 27 | *Sunflower Student Movement* | 47 |
| 13 | *Josh Shapiro* | 69 | 28 | *The Battles of Coxinga* | 47 |
| 14 | *Laura Siegemund* | 61 | 29 | *Transportation in Washington, D.C.* | 71 |
| 15 | *Long QT syndrome* | 55 | 30 | *Zhejiang University* | 61 |

Table 5: Domains that are used for generating new queries. #: the count of valid queries in our final (1,835) data.

For the test annotation, we require that their accuracy in determining the correctness of GPT-4 responses should reach at least 90%, and 90% of their annotation on gold responses should be reasonable and correct to the queries.

After annotation, we conduct a quality control process where we randomly select 10% of the annotations for manual review. If we discover that more than 5% of an annotator's submissions are deemed unsatisfactory, we request the annotator to redo the entire set of annotations to meet our quality standards.

### C.3 Annotator Background and Compensation

All annotators are junior undergraduate students who study in English programs, ensuring they have a solid foundation in reasoning and language necessary for the task. Annotators are compensated by payment based on their annotation quantity, which is around 1.4 USD per instance. This compensation structure and rate is applied across both the annotation of human-in-loop feedback and the construction of the SC-G4 benchmark.

### C.4 Estimated Human Performance on SC-G4

We provide three estimated human performance (binary-label) on SC-G4 benchmark. (1) We measure untrained annotators' performance by calculating their first trial annotation on 20 instances. 5 annotators were involved (1 did not pass our follow-up test annotation). Their Fleiss' Kappa score is 0.56 (moderate agreement). The average accuracy by majority vote is 81.00%. (2) We measure their trained performance by evaluating their test annotation on 100 instances, where we provide gold labels. Their averaged accuracies are 90.20% (w 1 who did not pass) and 92.50% (wo), respectively. (3) We measure their formal annotation performance by comparing their first round of formal annotation, and the final data (gold) after checking and revision. The accuracy is 96.73%.

## D   Seed Instance Collection

We collect those failure cases from two main sources: (1) papers that investigate the GPT-4 failure cases (Zheng et al., 2023b; Borji, 2023; Qin et al., 2023; Tan et al., 2023a) The original sources of those papers include HotPotQA (Yang et al., 2018), TSQA (Li et al., 2021), NQ (Kwiatkowski et al., 2019), BigBenchHard (Suzgun et al., 2022), BoolQ (Clark et al., 2019). (2) failure cases from online (twitter) and our usage. For example:

> *"If knaves always lie and knights always tell the truth, and you ask a person if they are a knight and they say no, what are they?"*

For cases presented via screenshots, we manually convert them into text. All data are then tested using our GPT-4 API.

# E    Full Domain Information

Table 5 presents the full domains that we use to generate new queries in the SC-G4 benchmark and the number of valid queries of individual domains. These domains are Wikipedia titles, which are randomly selected from Wikipedia metadata (2023).

# F    GPT-4 Error Patterns

We present the full GPT-4 error patterns, with each coupled with their ID and initial patterns for comparison. We show the major difference in *italics*.

## F.1    Pattern A

**Initial Pattern A**

> Assumption of Existence: GPT-4 may generate incorrect answers when a question assumes the existence of a non-existent entity or concept. This can lead the model to provide an answer based on incorrect assumptions or associations. Such entities or concepts are created using the combination of existing ones.
>
> To generate questions containing this pattern, create the existence of a non-existent entity or concept by slightly modifying existent entities, or combining multiple existent entities into one, and then ask questions that assume the existence of such non-existent entities or concepts within a real context.

**Final Pattern A**

> Assumption of Existence: GPT-4 may generate incorrect answers when a question assumes the existence of a non-existent entity or concept. This can lead the model to provide an answer based on incorrect assumptions or associations. Such entities or concepts are created using the combination of existing ones.
>
> To generate questions containing this pattern, create the existence of a non-existent entity or concept by slightly modifying existent entities, or combining multiple existent entities into one, and then ask questions that assume the existence of such non-existent entities or concepts within a real context. *Additionally, make the questions more specific and detailed, so that GPT-4 is more likely to generate an answer based on the assumed existence of the non-existent entity or concept, rather than recognizing its non-existence. For example, instead of asking about the impact of a non-existent study, ask about specific findings or methodologies used in the study. This will make it more challenging for GPT-4 to recognize the non-existence of the study and may lead to incorrect answers.*

## F.2    Pattern B

**Initial Pattern B**

> Bias towards More popular or Well-known Information and Overgeneralization: GPT-4 might be more familiar with popular or well-known topics

due to their frequency in the training data. This could lead the model to generate answers related to more popular topics, even if they are incorrect. Also, GPT-4 might overgeneralize from its knowledge of a topic, leading it to choose a more well-known or common answer instead of the correct one.

To generate questions with this pattern, create questions that involve less well-known aspects of popular topics or require the AI to differentiate between popular and less popular information. To generate questions with overgeneralization, create questions that require the AI to avoid common assumptions or generalizations and focus on specific details or less well-known aspects of a topic. Or focus on subjects or situations that share similarities with other, more well-known examples, but have distinct differences that GPT-4 might overlook.

**Final Pattern B**

Bias towards More popular or Well-known Information and Overgeneralization: GPT-4 might be more familiar with popular or well-known topics due to their frequency in the training data. This could lead the model to generate answers related to more popular topics, even if they are incorrect. Also, GPT-4 might overgeneralize from its knowledge of a topic, leading it to choose a more well-known or common answer instead of the correct one.

To generate questions with this pattern, create questions that:

*1. Involve less well-known aspects of popular topics or require the AI to differentiate between popular and less popular information, but also ensure that the information is not outdated. And avoid asking questions with words like "unsung" or "lesser well-known".*

2. Require the AI to avoid common assumptions or generalizations and focus on specific details or less well-known aspects of a topic.

3. Focus on subjects or situations that share similarities with other, more well-known examples, but have distinct differences that GPT-4 might overlook.

*4. Involve information that is not easily accessible or requires a deeper understanding of the subject matter.*

*5. Avoid questions that can be answered by general knowledge or information that is widely available*

Ensure that questions generated by the new pattern are answerable and can be answered by a short phrase

### F.3 Pattern C

**Initial Pattern C**

Complex Counting or Identification Tasks: GPT-4 may struggle with counting tasks, leading to incorrect answers. Examples: "How many words in the following sentence have an odd number of letters?", "How many characters are in this sentence, not counting spaces and punctuation?", and "How many words in this sentence contain exactly three vowels?"

GPT-4 may struggle with accurately identifying specific letters or characters within words. Example: "How many words in the following sentence start with a consonant?"

**Final Pattern C**

Complex Counting or Identification Tasks: GPT-4 *often* struggles with accurately counting or identifying specific elements in complex sentences or lists. This can lead to incorrect answers when the question *requires counting words, characters, or instances of a specific criterion in a more intricate context.*

*Create questions that require GPT-4 to count or identify specific elements in complex sentences or lists. This can include tasks like counting words with a specific number of letters in a long sentence, identifying words that start with a specific letter in a paragraph or that contain a specific letter at specific position, or counting characters in a sentence with multiple punctuation marks and special characters. Avoid simple and straightforward questions that GPT-4 can easily answer correctly.*

### F.4 Pattern D

**Initial Pattern D**

Complex Logical Reasoning and Paradoxes: GPT-4 may not be able to handle questions involving paradoxes or complex logical reasoning. Example: "If knaves always lie and knights always tell the truth, and you ask a person if they are a knight and they say no, what are they?"

**Final Pattern D**

Complex Logical Reasoning and Paradoxes: GPT-4 may struggle with questions involving complex logical reasoning, hypothetical scenarios with multiple variables, or paradoxes. *These questions often require a more sophisticated understanding of logic and reasoning than the model is capable of handling.*
Example: "If knaves always lie and knights always tell the truth, and you ask a person if they are a knight and they say no, what are they?"
- Focus on questions that involve multiple variables and logical steps that need to be connected in order to arrive at a correct answer.
*Example: "Imagine a scenario where the Edinburgh and Northern Railway has a rule that trains can only stop at stations with prime-numbered platforms, and every station has a platform number that is doubled each time a train arrives. If a train starts at platform 1 and continues to the next station with a prime-numbered platform, what would be the platform number of the 4th station it stops at?"*
*- Create hypothetical scenarios that involve conditions or constraints that make the question more challenging to answer.*
*- Include paradoxical elements or contradictions in the question that make it difficult to provide a straightforward answer.*
*- Avoid questions that can be answered with a general analysis or by providing relevant information about a specific topic, as GPT-4 is likely to answer these correctly.*

### F.5 Pattern E

**Initial Pattern E**

Complex Syntactic Structures and Multiple Clauses: GPT-4 struggles to accurately parse sentences with complex structures and multiple clauses, leading to incorrect identification of specific words or relationships between words.
To generate questions that challenge GPT-4, create sentences with multiple clauses, embedded phrases, and complex relationships between words and ask GPT-4 to parse the sentence or identify specific words or relationships between words. This will make it difficult for GPT-4 to parse the sentence accurately and identify the correct answer

**Final Pattern E**

Complex Syntactic Structures and Multiple Clauses: GPT-4 struggles to accurately parse sentences with complex structures and multiple clauses,

leading to incorrect identification of specific words or relationships between words.

To generate questions that challenge GPT-4, create long and complex sentences with multiple clauses, embedded phrases, and complex relationships *(such as dependency or constituency) between words. Then, ask GPT-4 to parse the sentence, identify specific words or dependency or constituency relationships between words, or rephrase the sentence.*

*For example: 1. "I'm a student very close to the VCU area that's looking for an apartment or house to move in to with another person/people." What are the words in the previous sentence, which form a constituency with the second word that begins with letter 'a'?*

*2. "Unfortunately, I currently have no funding or capacity to advise interns or visiting students. Please consult the EdiNLP web site for more information on potential supervisors." What are the words in the previous sentence, which hold a dependency with "funding" or what are the predicative words in the above sentences?*

*This will make it difficult for GPT-4 to parse the sentence accurately and identify the correct answer. Avoid fact-based questions that do not require complex linguistic analysis.*

### F.6 Pattern F

**Initial Pattern F**

Recursive and Unusual Patterns: Questions that require recursive operations, such as replacing characters within a word multiple times, can be challenging for GPT-4. Example: "What is the outcome of replacing all the 't's in the word 'tomato' with 'potato'?"

GPT-4 might struggle with questions involving unusual patterns or tasks that require non-standard processing. Examples: "How many words in the following sentence have an odd number of letters?", "How many words in this sentence contain exactly three vowels?", and "How far is Aigre located north of Angoulême?"

**Final Pattern F**

Recursive and Unusual Patterns: GPT-4 might struggle with questions involving unusual patterns or tasks that require non-standard processing, such as recursive replacements or unconventional counting tasks.

*a. Increase the complexity of the unusual patterns or tasks, such as incorporating more steps or multi-level replacements in the questions.*

*b. Combine multiple unusual patterns or tasks within a single question to increase the difficulty.*

*c. Focus on generating questions that require unconventional counting tasks or identifying patterns across multiple words or phrases.*

*By increasing the complexity and incorporating multiple unusual patterns or tasks within a single question, we can generate questions that are more likely to induce GPT-4 to generate incorrect answers. However, it is essential to ensure that the generated questions remain answerable and can be answered by a short phrase.*

### F.7 Pattern G

**Initial Pattern G**

Temporal Ambiguity or Confusion with Specific Events: GPT-4 might have difficulty understanding the significance of specific time frames mentioned in the question and could provide information about the subject's career

or life in general, rather than focusing on the specific time frame. To generate questions with this pattern, include precise time frames (e.g., specific months or years) that might be challenging for GPT-4 to pinpoint accurately.

**Final Pattern G**

Temporal Ambiguity or Confusion with Specific Events: GPT-4 might have difficulty understanding the significance of specific time frames mentioned in the question and could provide information about the subject's career or life in general, rather than focusing on the specific time frame. To generate questions with this pattern, include precise time frames (e.g., specific months or years. *But do not be too specific, e.g., specific dates or hours) that might be challenging for GPT-4 to pinpoint accurately, and also mention specific events or accomplishments that occurred during those time frames. This will make it more difficult for GPT-4 to provide a correct answer without addressing the specific event or accomplishment.*

### F.8  Pattern H

**Initial Pattern H**

Text Manipulation or Transformation: GPT-4 can make errors when performing text manipulation or transformation tasks, such as replacing letters, shifting letters in the alphabet, or sorting words. These errors can be due to the complexity of the task or limitations in GPT-4's processing capabilities.

Create questions that require GPT-4 to perform complex text manipulation or transformation tasks, such as replacing specific letters with others, shifting letters in the alphabet, or sorting words alphabetically.

**Final Pattern H**

Text Manipulation or Transformation: GPT-4 can make errors when performing complex text manipulation or transformation tasks, such as replacing specific letters with others based on certain conditions, shifting letters in the alphabet by varying amounts, or sorting words based on multiple criteria. These errors can be due to the complexity of the task or limitations in GPT-4's processing capabilities.

Create questions that require GPT-4 to perform complex text manipulation or transformation tasks, such as:

- Replacing specific letters with others based on their position in the word or the alphabet.

*- Shifting letters in the alphabet by different amounts depending on their position in the word or other conditions.*

- Sorting words based on *multiple criteria*, such as alphabetically and *by length*.

*Avoid questions that involve simple text manipulation tasks, such as sorting words alphabetically or replacing a single letter with another. To increase the difficulty, ask questions that combine multiple manipulation or transformation.*

## G  Self-Challenge Llama-3-70B

We apply our framework to Llama-3-70B (Dubey et al., 2024) and discover 19 patterns (due to limited space, we only show the pattern names) as shown in Table 6.

However, we find that (1) those patterns are mostly covered by GPT-4. For example, GPT-4 pattern B covers Llama patterns H and J, GPT-4 pattern E covers Llama patterns O, Q and S. (2) some GPT-4 patterns are not found by Llama3. For example, GPT-4 patterns C

| ID | Pattern Name |
|----|--------------|
| L-A | Misleading context with a provided date |
| L-B | Irrelevant information with a date |
| L-C | Lack of explicit current date |
| L-D | Inconsistent or implicit information |
| L-E | Assumption of Existence |
| L-F | Ambiguity and Vagueness |
| L-G | Lack of Context |
| L-H | Overfitting and Bias |
| L-I | Misinterpretation of Syntax and Semantics |
| L-J | Inability to Handle Novel or Unseen Data |
| L-K | Misinterpretation of Context |
| L-L | Incomplete or Partial Analysis |
| L-M | Misunderstanding of Instructions |
| L-N | Limited Knowledge or Training Data |
| L-O | Linguistic Complexity |
| L-P | Format or Presentation Errors |
| L-Q | Complex Sentence Structures |
| L-R | Pronoun Resolution |
| L-S | Dependency Identification |

Table 6: A list of names of patterns discovered by Llama-3-70B.

(*Complex Counting or Identification Tasks*), D (*Complex Logical Reasoning and Paradoxes*), H (*Text Manipulation or Transformation*).

We further optimize Llama pattern E. We then find Llama-3-70B cannot generate queries that induce Llama-3-70B to hallucinate.

## H   Examples of New Queries

Table 7 presents the examples of new queries for each pattern. We see that those new queries contain the features of corresponding patterns, and are diverse in domains.

## I   Evaluation Validation

To validate the effectiveness, we first compare the gold labels and model-evaluate labels of GPT-4 responses on the full 1,835 instances, where GPT-4 evaluation achieves 85.35 precision, 95.89 recall and 90.31 *F*-1. As GPT-4 evaluation can be biased to GPT-4 output (Liu et al., 2023), we further construct a small set that consists of 115 instances, where each gold response is completely annotated by human annotators without the assistance of GPT-4. The GPT-4 evaluation achieves 87.14 precision, 96.83 recall and 91.73 *F*-1 on this set, which we consider is good enough. Such a higher-recall and lower-precision evaluation indicates that GPT-4 is highly reliable in identifying correct answers but with a few errors in classifying incorrect responses as "*correct*". It also suggests that the model performance evaluated by GPT-4 can be slightly overestimated or higher than its real performance.

We also experiment with the evaluation method following Wang et al. (2023), i.e., asking LLMs to explain why the model-generated response is correct or not, and then give a label for the response. However, we do not observe significant improvement.

## J   Analyses of Model Performance on Different Domains

Figure 3 shows the breakdown of model performance on different domains. Generally, GPT-4 shows the best performance on all domains, and gives the most smooth curve compared with other models. In contrast, Llama 2 models show worse performance, and give more sharp curves, indicating they are less stable and generalizable across domains, compared with GPT-4 and Turbo.

| ID | Pattern Name | Example |
|---|---|---|
| A | Assumption of Existence | During the "Karpov Invitational" tournament, what unique rule did Anatoly Karpov introduce to encourage faster and more aggressive play? |
| B | Bias towards More popular or Well-known Information and Overgeneralization | In the context of the Philippine drug war, which specific law enforcement unit was disbanded in 2017 due to its controversial involvement in extrajudicial killings? |
| C | Complex Counting or Identification Tasks | In the sentence, "Joey Votto is known for his exceptional plate discipline and ability to get on base," how many words have the letter 'e' as the third character and end with the letter 'n'? |
| D | Complex Logical Reasoning and Paradoxes | If Muirchertach Ua Briain had a rule that he could only levy taxes on years with a prime number of battles, and the number of battles doubled each year, which year would be the fourth time he levied taxes if his first tax year had 5 battles? |
| E | Complex Syntactic Structures and Multiple Clauses | The entrance of Abney Park Cemetery features elaborate iron gates, designed by the architect William Hosking, that exhibit the Victorian Gothic architectural style. What are the words in the previous sentence that hold a dependency relationship with "iron gates"? |
| F | Recursive and Unusual Patterns | If the words "rapid," "accurate," and "scalable" are used to describe the ideal characteristics of a COVID-19 test, and you were to arrange these words based on the sum of the positions of their vowels in the alphabet, which word would come first? |
| G | Temporal Ambiguity or Confusion with Specific Events | During the fall of 1989, Ray Cooper participated in a concert tour with a famous rock band. What was the name of that band? |
| H | Text Manipulation or Transformation | In the context of Martin Nolan's work, replace the first and last letters of each word in the phrase "auction house executive director" with their corresponding opposite letters in the alphabet (A <- >Z, B <- >Y, etc.), and then sort the transformed words by length in descending order. |

Table 7: Examples of new queries generated by GPT-4.

|  | Train | Dev | Test |
|---|---|---|---|
| Count of instances | 500 | 325 | 1,010 |
| Avg Query | 34.22 | 36.94 | 4.48 |
| Avg Gold R | 61.61 | 63.67 | 38.19 |
| Avg GPT-4 R | 82.40 | 84.50 | 73.97 |

Table 8: Data statistics for fine-tuning data split by difficulty. Avg: averaged token length. R: Response.

## K   Analysis of Few-shot Performance on Data Split by Difficulty

We investigate whether fine-tuned models can be improved on data that is most challenging to GPT-4. In particular, we take all instances where GPT-4 fails (human evaluation) as test set (1,010), and randomly split the rest data into train (500) and dev sets (325). The statistic is shown in Table 8.

Table 9 shows the results. First, we see that GPT-4 evaluation can highly overestimate the model performance on this most challenging test set compared with on other test sets. This can be because that those instances are very to challenge to GPT-4 itself in terms of diverse

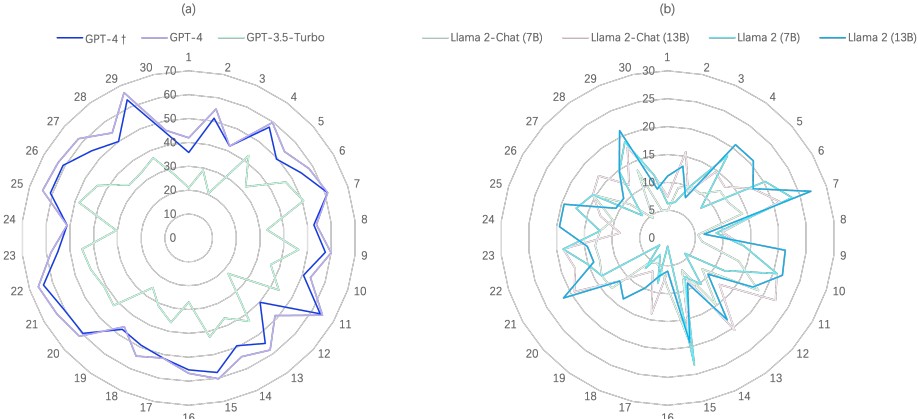

Figure 3: Breakdown analysis of zero-shot performance under different individual domains. The IDs of domains are corresponding to IDs in Table 5 in Appendix E, respectively.

| ID | # | G-4$^\dagger$ | G-4 | L-2 | L-2$^*$ |
|---|---|---|---|---|---|
| A | 58 | 0.00 | 10.34 | 1.72 | 29.31↑ |
| B | 76 | 0.00 | 13.16 | 2.63 | 3.95↑ |
| C | 180 | 0.00 | 3.33 | 1.11 | 5.00↑ |
| D | 73 | 0.00 | 15.07 | 1.37 | 1.37 |
| E | 109 | 0.00 | 9.17 | 5.50 | 6.42↑ |
| F | 134 | 0.00 | 10.45 | 0.75 | 1.49↑ |
| G | 134 | 0.00 | 9.96 | 2.24 | 7.46↑ |
| H | 246 | 0.00 | 2.85 | 1.22 | 0.00↓ |
| all | 1,010 | 0.00 | 7.52 | 1.81 | 4.95↑ |

Table 9: Model performance on data split by difficulty.

factors, including their internal knowledge, and GPT-4 can fail to detect errors in model response, and thus be more biased.

## L  Analyses on Pattern A

Pattern A finds that GPT-4 tends to hallucinate when the input queries contain non-existent entities, which are mostly created by combining several existent entities into one. Interestingly, fine-tuned models show the most improvement on this pattern, thus we manually investigate further by looking into the model outputs.

Overall, we find that fine-tuned Llama 2 gets higher accuracy on such a pattern because it has learnt to reject to answer ( *"There is no ..."*).

Take the below case for example:

> **Query**: *How did the Edinburgh and Northern Railway's experimental solar-powered train perform in terms of energy efficiency during its trial run in 1858?*

> **Gold**: *There was no solar-powered train in 1858 actually. But the first solar cell was invented in 1883 by Charles Fritts.*

> **GPT-4 (zero)**: *I'm sorry, but there seems to be a mix-up in the information you provided. The Edinburgh and Northern Railway was a railway company in Scotland that operated from 1847 to 1862. Solar power technology was not available during that time period, and the first solar cell was invented in 1954 by*

| ID | Dist-3 | Dist-4 |
|---|---|---|
| A | 87.02 | 93.93 |
| B | 87.71 | 94.42 |
| C | 74.56 | 84.33 |
| D | 81.68 | 90.76 |
| E | 82.03 | 88.74 |
| F | 75.30 | 86.65 |
| G | 82.62 | 92.00 |
| H | 46.19 | 59.41 |
| all | 70.23 | 82.05 |

Table 10: Distinct scores of SC-G4.

| word sorting | Dist-3 | Dist-4 |
|---|---|---|
| wo. prefix | 41.63 | 49.39 |
| w. prefix | 8.72 | 9.98 |

Table 11: Distinct scores of BigBench word-sorting task.

*Bell Labs. Therefore, it is not possible for the Edinburgh and Northern Railway to have had an experimental solar-powered train in 1858.* (correct)

**Llama 2-Chat (zero)**: ***The Edinburgh and Northern Railway's experimental solar-powered train, which ran in 1858, was able to achieve an energy efficiency of around 15%.*** *This means that for every unit of energy used to power the train, it was able to travel approximately 15 units of distance. While this may not seem particularly impressive by modern standards...* (incorrect)

**Llama 2-Chat (fine-tuned on data randomly-split)**: *There is no record of an Edinburgh and Northern Railway's experimental solar-powered train trial run in 1858.* (correct)

**Llama 2-Chat (fine-tuned on data spilt by domain)**: *Thank you for your question!* ***However, I must point out that there is no record of an Edinburgh and Northern Railway's experimental solar-powered train in 1858....*** *(correct)*

we see that the fine-tuned Llamas successfully identify that this query can contain non-factual content, which is considered correct, but it fails to respond informatively as GPT-4 and human. We further investigate the outputs of Llama-2-Chat (7B) fine-tuned on randomly split data, and find that over 50% of queries under pattern A are rejected in a similar way by it.

Moreover, it is noticeable that the cross-domain fine-tuned Llama (domain split) also successfully rejects such questions. Recall that in this setting, we constrain the domains in the training and testing sets. In other words, pattern knowledge is shareable while domain knowledge is not. The results indicate that the fine-tuned Llama effectively learns the hidden features of these patterns and can apply them across different domains. We hypothesize that it is possible that LLMs can sense such uncertainty within themselves (also note the uncertainty can exist across domains). By fine-tuning on such data, LLMs can be better at distinguishing between factual and certain entities and non-factual ones, and refuse to answer questions with non-existent entities or events, rather than generating non-factual responses.

## M  Diversity Evaluation of SC-G4 Queries

We use the Distinct score (Li et al., 2015) to evaluate the diversity of generated queries. We report dist-3 and dist-4 in Table 10. Higher scores indicate the data are more diverse. For comparison, we show the results of the BigBench word-sorting task (bench authors, 2023) in Table 11, which is similar to our pattern H. Generally, we see that queries in SC-G4 are diverse from each other.

