# OpenReview forum: "See What LLMs Cannot Answer: A Self-Challenge Framework for Uncovering LLM Weaknesses"
_colmweb.org/COLM/2024/Conference — COLM_

### Official Review · Reviewer_oXPQ · 2024-04-12

**Rating:** 6
**Confidence:** 4
**Ethics Flag:** 1

**Summary:**

This paper proposes a framework to analyze error patterns in mistakes made by LLMs by using analysis by LLMs and feedback by humans. By using the proposed framework, the experiment identifies 8 patterns of errors made by GPT-4. In addition, they introduce a benchmark that includes new queries in the 8 patterns. The analysis on the benchmark shows that GPT-4 makes mistakes in more than 50% of cases and fine-tuning cannot solve the benchmark.

**Questions To Authors:**

(comments and suggestions)
* Word sorting is not a novel task. For example, this task is in Big-Bench (https://github.com/google/BIG-bench/tree/main/bigbench/benchmark_tasks/word_sorting).
* Step 3 in Figure 1 does not explain almost anything about pattern optimization. I recommend to add more details.
* I recommend providing the versions of the GPT models you used (e.g., 0613) to improve reproducibility.

**Reasons To Accept:**

This paper proposes an idea for analyzing mistakes made by LLMs using LLMs. This idea provides a relatively reproducible framework for diagnosing problems in responses from LLMs, beyond evaluating performance such as accuracy on specific tasks. This framework may help future research in creating new LLMs or tuning LLMs for specific applications.

**Reasons To Reject:**

The main problem with this paper is the lack of information about the Seed Instance Collection (Section 3.1). This paper should at least clearly show the sources of 189 queries, and I recommend providing the full list of the actual queries in the appendix or supplemental materials. If details of these seed instances are not provided, all experiments in this paper provide no information to readers. (The other issues below are much more minor. Please first focus on addressing this problem.)

In addition, it looks like this paper only evaluates tasks in limited domains (factual questions and logical tasks like word sorting). If this is true (I'm not sure because details are not provided), this paper should clearly mention that many other popular tasks, such as arithmetic reasoning or text generation, are not covered.

The experiments in this paper are insufficient to claim that the observations apply to diverse LLMs. (1) Although this paper analyzes other LLMs on the proposed benchmark, it mostly only focuses on GPT-4. It is better to show whether we see different patterns when the proposed framework (in Figure 1) is applied to other LLMs. (2) Experiments in Section 5 only evaluate GPT and small Llama, which are insufficient. I recommend evaluating at least one more strong closed model (e.g., Gemini, Claude) and a large open model (e.g., Mixtral, Llama 70B).

(Updated June 5th) I've updated the overall rating from 5 to 6 since the rebuttal by the authors partially addresses my concerns.

---

> ### Author Rebuttal · Authors · 2024-05-31
>
> Thank you for your valuable review.
>
> **Seed Instance Collection**
>
> The full list can be found at https://anonymous.4open.science/r/Self_chal_anonymous-9205/seed_instance.json.
> In this research, we are motivated by the failure of GPT4, and we collect those failure cases from:
>
> (1) 4 papers that study the failure of ChatGPT(4) [1,2,3,4] (67.7%), where we focus on QA data.
>
> 1. Why Does ChatGPT Fall Short in Providing Truthful Answers?
> 2. A Categorical Archive of ChatGPT Failures
> 3. Is ChatGPT a General-Purpose Natural Language Processing Task Solver?
> 4. Towards Benchmarking and Improving the Temporal Reasoning Capability of LLMs
>
> For example:
>
> ```The school in which the Wilmslow Show is held is designated as what?```
>
> ```Who did Jessica Valenti work for in 2005?```
>
> The original sources of those papers include HotPotQA, TSQA, NQ, BigBenchHard, BoolQA.
>
> (2) failure cases from online (twitter) and our usage (32.3%). For example:
>
> ```If knaves always lie and knights always tell the truth, and you ask a person if they are a knight and they say no, what are they?```
>
> For cases presented via screenshot, we manually convert them into text. All data are then tested using our GPT-4 API.
>
> We will further clarify those sources and provide full seed instances in our revision.
>
> **Evaluation in Limited Tasks**
>
> First, we acknowledge that we did not involve traditional NLG tasks, e.g. MT, due to the fact as you mention. We will discuss it as a limitation.
>
> Second, 8 tasks (patterns) are summarized from 189 data. With more data collected, we believe there would be more interesting tasks (patterns).
> In fact, the Self-Chal benchmark involves arithmetic reasoning tasks such as Patterns C and D. For example:
>
> ```... If a student starts with a paper containing 5 grid squares and each subsequent paper's number of squares is doubled, what will be the total number of characters written on the 4th paper?```
>
> **Using More LLMs for Experiments**
>
> We use Llama3-70B-instruct, which summarizes 21 patterns. However, we find Llama3 is less effective compared with GPT4: (1) one G-4 pattern can cover multiple L-3 pattern; (2) L-3 patterns cannot be used to generate challenging queries; (3) multiple G-4 patterns are not shown in L-3 patterns.
>
> We benchmark more LLMs on Self-Chal. Please also see Response to xmpu.
>
> **Suggestion on clarification**
>
> We will follow your suggestion to revise our paper. We use gpt-4-32k-0613.
>
> We would really appreciate it if you could reconsider our paper.

---

> > ### Comment · Reviewer_oXPQ · 2024-06-05
> > **Re: Rebuttal by Authors**
> >
> > Thank you for your response. I appreciate you providing the details of the seed instances.
> >
> > I still believe that the experiments should be conducted on more diverse data and models, as you mentioned as a limitation. However, since my main concern was partially addressed, I've updated the overall rating from 5 to 6.
> >
> > I expect that more details about the seed instances will be explained in future versions of your paper.

---

> > > ### Author Response · Authors · 2024-06-05
> > > **Response to Reviewer's comment**
> > >
> > > We really appreciate your response and thank you for increasing the score!
> > >
> > > **Regarding the experiments with more data and models**
> > > Yes, we acknowledge that Self-challenging LLMs with more seed data would be beneficial.
> > >
> > > We will include 1) our new experiments using Llama-3-70B-instruct to self-challenge; 2) new results that evaluate more diverse LLMs on Self-Chal data (including Gemma, Phi-3, Mixtral, Llama3, Gemini, Claude and GPT-4o) (as presented in our reponse to xmpu: https://openreview.net/forum?id=18iNTRPx8c&noteId=wQL6qdwBWh) in our revision.
> > >
> > > We will add all above clarification in our revision.

---

> ### Author Response · Authors · 2024-06-05
> **Discussion**
>
> Due to the limited space (2500 characters) for rebuttal, we respond to the main questions. Here, we take the opportunity to engage further discussion.
>
> **Figure 1 clarification**
>
> We have modified the Figure 1: https://anonymous.4open.science/r/Self_chal_anonymous-9205/Figure_1.pdf (please click View raw to see) and will add a clarification on the caption as below.
>
> ```
> Figure 1:  The overall Self-Challenge framework. We first summarize initial error patterns from seed failure instances (Step 1). Then, we perform pattern evaluation (Step 2) to evaluate the quality of summarized patterns, and obtain corresponding human feedback; pattern optimization (Step 3) to modify the original pattern based on human feedback. We frame Step 2 and Step 3 iteratively. We present the difference between the Initial Pattern and the Optimized Pattern in underlined text.
> ```
>
> **Word sorting**
>
> Thank you for your suggestion. We will clarify it in our revision.
>
> We hope that our answers have been satisfactory and that we've addressed your concerns effectively.
> Please let us know if you have any further questions or require clarification on any points. We are happy to engage in further discussion.

---

### Official Review · Reviewer_xmpu · 2024-05-10

**Rating:** 5
**Confidence:** 4
**Ethics Flag:** 1

**Summary:**

The authors propose a Self-Challenge (Self-Chal) evaluation framework designed to uncover weaknesses in LLMs by having LLMs and humans collaboratively identify challenging patterns and generate new instances based on those patterns. They start with seed instances that GPT-4 struggles to answer, prompting GPT-4 to summarize error patterns. These patterns are used to generate new instances, which are iteratively refined with human feedback. The authors present eight patterns, such as text manipulation and questions with assumptions, and create a benchmark, Self-Chal, consisting of 1,835 instances with human-annotated gold responses. Their results show that GPT-4 can correctly answer 44.96% of the Self-Chal instances. They find that these patterns cannot be fully resolved through fine-tuning, indicating that some error patterns may represent inherent flaws in LLMs.

**Questions To Authors:**

- The paper introduces the initial list of error patterns derived from GPT-4’s performance. What methodology was used to inform the initial pattern list? Was there a comprehensive and systematic approach, or was it primarily exploratory? Also, are the 8 error patterns (derived after two iterations of refinement) exhaustive in identifying all possible areas where GPT-4 fails, or are there additional patterns that could have been explored?

- I noticed that the prompts (e.g., the summarization prompt in B.1) are quite intricate and have "disclaimers" and warnings for the model to very specifically guide it/steer it away from certain behaviors. How did this prompt evolve over the iterations, and were earlier prompts less effective in summarizing patterns? If so,

- Why was GPT-4 specifically chosen for constructing the benchmark, and could other LLMs have been equally effective in generating challenging questions? Does the use of GPT-4 to evaluate its own responses lead to inherent bias in the evaluation, and how could this be mitigated? If there were iterative changes to the prompts, how can you ensure that the final prompts are not simply optimized for gaming the results in your experiments rather than revealing genuine weaknesses in GPT-4 and other LLMs?

- I have noticed that particularly GPT-4 tends to become overly focused on the unimportant specifics of provided examples/details in a prompt, leading to generated instances that are not diverse. How did you address this issue, which I fear might lead to instances in your benchmark that lack diversity and are too similar to one another? If you did combat against it, is there a way you can demonstrate the diversity of the uniqueness of queries in your dataset? On that note, it would be quite important to have a table of examples of instances in the paper so that the reader can "see for themselves" the types of instances.

**Reasons To Accept:**

- The Self-Challenge framework offers a relatively unique way to categorize weaknesses in LLMs according to interesting categories through an iterative protocol that harnesses LLMs themselves, rather than relying solely on accuracy. There are some recent similar efforts in the same spirit, e.g., The Generative AI Paradox in Evaluation: "What It Can Solve, It May Not Evaluate" (Oh et al, 2024) and WSC+: Enhancing The Winograd Schema Challenge Using Tree-of-Experts (Zahraei and Emami, 2024), which the authors should compare/contrast with in Related Work.

- The paper includes a commendable assessment of LLMs using multiple strategies beyond zero-shot, including few-shot and fine-tuning experiments.

**Reasons To Reject:**

**1. Lack of Comprehensive Comparative Analysis:**
- The paper relies exclusively on GPT-4 for error pattern generation, missing comparisons with other LLMs during the pattern discovery process. This is particularly important, since the idea of their challenge is to pit LLMs against their own recognitions of weaknesses, then LLM-diversity in the generated dataset is crucial.
- The evaluative experiments include a limited set of models (primarily GPT and LLaMA families), neglecting many other available LLMs, both open-source and proprietary (e.g., Mixtral, Gemma, Phi, Claude 3) that could provide a more comprehensive understanding of error patterns across models.  Because of this, the claim that error patterns are "universal" (which the authors often make) across LLMs is too strong given the limited evaluation on just a few families of models.
- The training data for fine-tuning is too small to draw general conclusions about the impact of fine-tuning on model improvement (although the authors acknowledge this).

**2. Inadequate Experimental Depth:**
- The experiments lack a proper comparison to human performance on the Self-Chal benchmark.
- The results focus only on GPT-4-generated questions without sufficient discussion on potential biases.
- Qualitative error analyses would be very important to enhance the depth of the experiments.
- Although the evaluation protocol was validated with human labels, the validation set was relatively small, which could lead to inaccuracies in automated evaluation.
- The paper does not provide a baseline comparison with human performance on the Self-Chal benchmark. Of course, I know that humans were involved in the labelling, but if so, then at the very least,a "majority vote" (supported with inter-annotator agreement statistics) using multiple annotators can used to proxy a "human baseline".

**3. Insufficient Coverage in the Related Work Section:**
- The related work section is too brief and currently resides in the appendix, which is problematic (particularly because there are many relevant works that need to be discussed as a matter that is of high priority as the main content of the paper). For example, it lacks coverage of a significant body of literature concerning dataset augmentation and creation using LLMs.

---

> ### Author Rebuttal · Authors · 2024-05-31
>
> Thank you for your valuable review.
>
> **Self-Challenging of Llama3-70B-instruct**
>
> The full patterns summarized by Llama3 can be found at https://anonymous.4open.science/r/Self_chal_anonymous-5FE3/README.md
>
> Generally, we find (1) Llama3 summarized patterns are mostly covered by GPT4. For example, P-B in our paper covers P-8 and 10, P-E covers P-15, 17 and 19. (2) some GPT4 patterns are not found by Llama3. For example, P-C (counting), P-D (logical), P-H (text manipulation).
>
> We further optimize P-5 and find Llama3 cannot generate queries that induce Llama3 to hallucinate.
>
> We will include full results in our revision.
>
> **Benchmarking other LLMs**
>
> |LLMs|A|B|C|D|E|F|G|H|all|
> |:-|:-:|:-:|:-:|:-:|:-:|:-:|:-:|:-:|:-:|
> |Gemma-7B|33.52|7.69|7.76|12.23|15.62|5.65|6.45|0|10.35|
> |Phi3-instruct-7B|12.29|20.36|14.22|35.37|20.70|22.61|7.83|2.21|16.84|
> | mixtral-8x7B | 24.02|25.79|7.33|19.65|26.17|13.48|7.83|0.37|15.15|
> |Llama3-70B-instruct|43.58|31.67|20.26|33.19|32.81|20.87|14.75|0.74|23.81|
> |Gemini-1-pro|22.91|16.82|7.79|16.81|25.78|10.09|7.01|0|13.06|
> |Claude-3-Sonnet|35.75|34.39|20.69|35.37|28.52|24.35|16.59|2.21|23.98|
> |GPT-4o|25.70|41.63|31.03|44.54|41.80|36.52|18.43|10.70|31.17|
>
> Generally, all LLMs show poor results. Similar trends can be consistent with previous research [1, 2] that LLMs can be homogeneous as they are trained with similar techs (e.g., transformers) on similar data (e.g., Wikipedia).
>
> 1. Picking on the Same Person: Does Algorithmic Monoculture lead to Outcome Homogenization?
> 2. The Platonic Representation Hypothesis
>
> **Human Performance**
>
> Due to space limit, please see response to nH6u, where we provide estimated human performance.
>
> **Evaluation Protocol**
>
> We acknowledge that the validation on 115 data is small. However, this is a supplement to our validation on full 1835 data.
>
> **Diversity**
>
> Following your suggestion, we evaluate the diversity of our data using Distinct score. We report dist-3 and dist-4 here. The higher, the more diverse.
>
> |group|dist-3|dist-4|
> |:-|:-:|:-:|
> |A|87.02|93.93|
> |B|87.71|94.42|
> |C|74.56|84.33|
> |D|81.68|90.76|
> |E|82.03|88.74|
> |F|75.30|86.65|
> |G|82.62|92.00|
> |H|46.19|59.41|
> |all|70.23|82.05|
>
> We compare BigBench word-sorting task, which is similar to pattern H.
>
> |word_sorting|Dist-3|Dist-4|
> |:-|:-:|:-:|
> |wo. prefix|41.63|49.39|
> |w. prefix|8.72|9.98|
>
> It shows our data are diverse and are of slightly higher diversity compared with similar task.
>
> We would really appreciate it if you could reconsider our paper.

---

> > ### Comment · Reviewer_xmpu · 2024-06-04
> >
> > Thanks for your response. Having gone over the other reviews, responses to them, as well as my own, I have decided to keep my scores as is.

---

> > > ### Author Response · Authors · 2024-06-05
> > > **Discussion**
> > >
> > > We appreciate your response. Due to the limited space (2500 characters), we respond to the main questions in our rebuttal. Here, we take the opportunity to further address your concerns.
> > >
> > > **Potential Biases of GPT-4 Generated Questions**
> > >
> > > Thank you for pointing out this issue. We will add the below clarification as a limitation.
> > >
> > > ```
> > > We acknowledge that relying on GPT-4 to generate instances for evaluation can lead to potential biases, in particular data homogeneity [1, 2]. Although we enrich the input pattern with additional domain information and evaluate their level of diversity by calculating the Distinct score [3], there still remains a potential problem that queries under the same pattern and domain can be similar to each other. This problem can be solved by decomposing the current patterns into more fine-grained and detailed sub-patterns, or providing larger domain text as a source for query generation, which we leave for future work.
> > > ```
> > >
> > > 1. Data Augmentation using LLMs: Data Perspectives, Learning Paradigms and Challenges
> > >
> > > 2. Diversify Your Vision Datasets with Automatic Diffusion-Based Augmentation
> > >
> > > 3. A diversity-promoting objective function for neural conversation models
> > >
> > >
> > > **More Related Work**
> > >
> > > We will add more discussion on recent studies, in particular those focusing on dataset augmentation and creation using LLMs. For example:
> > >
> > > ```
> > > Recently, researchers have explored using LLMs as annotators for creating and augmenting data of their interest, using both zero-shot and few-shot learning[1, 2]. For example, [3], [5], [9],[11], and [12] use LLMs such as ChatGPT and GPT-4 to label raw text for text classification tasks, and [6] and [13] use LLMs to generate output texts for text generation tasks, such as machine translation and text summarisation.  Additionally,  [4] finetunes GPT-2 with small data to annotate new training data for text classification tasks. [7], [8] and [10] use LLMs to augment dialogues with few-shot learning. Our work falls in the above studies that use LLMs for data augmentation. However, different from those that mostly use LLMs to augment analogous data or label output text, we use LLMs to obtain diverse queries for a specific task under the guidance of patterns (instruction) generated by LLMs themselves.
> > > ```
> > >
> > > Also, we thank you for the reference [14, 15]. We would add relevant discussion in our revision. For example, in Sec 4.4 (Evaluation protocol):
> > > ```
> > > As merely relying on GPT-4 to evaluate model response can be less effective [14, 15], we evaluate model performance by comparing their responses and gold responses.
> > > ```
> > > And in Appendix A:
> > > ```
> > > [14, 15] show that although LLMs can show superior performance compared with humans, they can have limitations in basic tasks.
> > > ```
> > > and
> > > ```
> > > [15] use LLMs to generate complex and multi-constrained instances, by drawing on common-sense reasoning. Compared with them, we focus on more general QA-format tasks.
> > > ```
> > >
> > > 1. Data Augmentation using LLMs: Data Perspectives, Learning Paradigms and Challenges
> > >
> > > 2. Large Language Models for Data Annotation: A Survey
> > >
> > > 3. AugGPT: Leveraging ChatGPT for Text Data Augmentation
> > >
> > > 4. Not Enough Data? Deep Learning to the Rescue!
> > >
> > > 5. Data Augmentation for Intent Classification with Off-the-shelf Large Language Models
> > >
> > > 6. Medically Aware GPT-3 as a Data Generator for Medical Dialogue Summarization
> > >
> > > 7. DIALOGIC: Controllable Dialogue Simulation with In-Context Learning
> > >
> > > 8. AugESC: Dialogue Augmentation with Large Language Models for Emotional Support Conversation
> > >
> > > 9. Selective In-Context Data Augmentation for Intent Detection using Pointwise V-Information
> > >
> > > 10. Tuning language models as training data generators for augmentation-enhanced few-shot learning
> > >
> > > 11. Chatgpt-4 outperforms experts and crowd workers in annotating political twitter messages with zero-shot learning.
> > >
> > > 12. Can ChatGPT Reproduce Human-Generated Labels? A Study of Social Computing Tasks
> > >
> > > 13. Prompting large language model for machine translation: A case study.
> > >
> > > 14. The Generative AI Paradox in Evaluation: "What It Can Solve, It May Not Evaluate"
> > >
> > > 15. WSC+: Enhancing The Winograd Schema Challenge Using Tree-of-Experts
> > >
> > >
> > > **Specific Prompt and Prompt Changes**
> > >
> > > In our early experiments, we find that GPT-4 did not behave as we expected. For example, we find that GPT-4 can summarize patterns that are very simple (e.g. "GPT-4 can fail on QA tasks") and do not contain detailed features. Thus, we add more specific instructions to guide GPT-4 to generate patterns that suit our goal: it should describe in what situations GPT-4 can fail, and it should be understandable by humans.
> > >
> > > However, once our prompts are set, we do not optimize them during the iteration of pattern evaluation (generating new queries) and optimization, i.e., ```there were no iterative changes to the prompts```.
> > >
> > > We hope that this discussion could address your concerns. Please let us know if you have any further questions and we are happy to engage in further discussion.

---

### Official Review · Reviewer_nH6u · 2024-05-11

**Rating:** 7
**Confidence:** 3
**Ethics Flag:** 1

**Summary:**

This paper presents Self-Chal, a self-challenging evaluation framework where LLMs summarize the error patterns (Step 1), evaluate the error patterns (Step 2), and optimize the patterns (Step 3). During this process, this study engages human experts to evaluate the quality of summarized patterns. This study obtains 8 patterns from 189 failure instances and constructs a new benchmark, Self-Chal, that consists of 1,835 challenging instances in the combination of 8 patterns and 30 domains. The experiments with multiple LLMs show that those error patterns are universal across different LLMs (e.g., Llama 2). In addition, fine-tuning LLMs cannot address these failure cases reliably.

**Questions To Authors:**

+ Where can I find the human performance on the new benchmark dataset (Self-Chal)? Is there any possibility that this method creates questions that cannot be solved by anyone?
+ It is interesting to see that LLMs suffer from answering questions about tokenization. At the same time, do we want LLMs to solve these questions (or "bugs" of LLMs) (from the practical point of view)?

**Reasons To Accept:**

+ This paper is well-written and easy to read.
+ The idea of letting LLMs analyze failure cases semi-automatically (with humans in the loop) and create difficult instances is interesting.
+ The new benchmark dataset (Self-Chal) may be useful for the development of LLMs.

**Reasons To Reject:**

+ The method presented in this paper is mostly about prompting LLMs (limited technical novelty); this may have pros and cons.

---

> ### Author Rebuttal · Authors · 2024-05-31
>
> Thank you for your valuable review.
>
> **Human Performance**
>
> We provide 3 estimated human performance (binary-label) on Self-Chal benchmark. (1) We measure untrained annotators’ performance by calculating their first trial annotation on 20 instances (before training). 5 annotators were involved (1 did not pass our follow-up test annotation). Their Fleiss' Kappa score is 0.56 (moderate agreement). The average acc by majority vote is 81.00%. (2) We measure their trained performance by evaluating their test annotation on 100 instances, where we provide gold-labels. Their averaged acc are 90.20% (w 1 who did not pass) and 92.50% (wo). (3) We measure 4 annotators' formal annotation performance by comparing their first round of formal annotation, and the final data (gold) after checking and revision. The acc is 96.73%.
>
> **Unanswerable Questions**
>
> Yes, there are questions that cannot be answered by one gold response. For example:
> ```Please combine two names of the centers and institutes at Stanford University in an alphabet order```.
>
> Such questions are filtered out of our benchmark, and are not used for evaluation.
>
> **Tokenization Issues**
>
> Thank you for acknowledging the tokenization tasks are interesting. From a practical view, such tasks can be closely related to applications such as poetry generation, spelling checking, and Teaching Aide in the area of NLP4Education. We also emphasize that such issues reveal the problems of underlying tokenization mechanisms, which can further influence the performance of downstream tasks. Also, it reveals that the current tokenization design in LLMs is different from humans.
>
> Lastly, we agree that our methodology mainly relies on prompting LLMs, which has been adapted by previous study [1, 2, 3, 4]. We also highlight that this framework is novel in that we propose a new semi-automatic evaluation protocol that can help humans better mine and understand the error patterns in LLMs.
>
> 1. NLG Evaluation using GPT-4 with Better Human Alignment
> 2. Better patching using LLM prompting, via Self-Consistency
> 3. Tree of thoughts: Deliberate problem solving with large language models
> 4. Chain-of-thought prompting elicits reasoning in large language models
>
> We thank you for your appreciation of our work.

---

> > ### Comment · Reviewer_nH6u · 2024-06-05
> > **Re: Rebuttal by Authors**
> >
> > Thank you for the response. I also agree with the issue of tokenization; LLMs are not aware of the nature of tokens that they are handling. I'm not sure from the response whether the human evaluation results were included in the current submission, but would recommend including this in the camera-ready version.

---

> > > ### Author Response · Authors · 2024-06-05
> > > **Response to Reviewer's Comment**
> > >
> > > We really appreciate your acknowledgement of our rebuttal.
> > >
> > > We will include human performance and all above clarification in our revision.

---

### Official Review · Reviewer_H8gk · 2024-05-12

**Rating:** 6
**Confidence:** 3
**Ethics Flag:** 1

**Summary:**

This paper introduces a self-challenge assessment framework that engages both LLM and humans to iteratively generate more challenging data and identify the limitations of LLM. This work is of the average quality, good clarity, average originality and good significance.

**Questions To Authors:**

1. Almost all answers are manually labeled, especially gold answers, which seems to contradict the claim of ‘self-challenge’. Suggested modification of this statement. If it must be manually labeled. Can the criteria for manual annotation be described in detail,. For example, the 5% annotation requirement for quality control is unsatisfactory, what is the standard weighing of unsatisfactory here?
 2.1 Regarding the optimization part, why did you choose to use GPT-4 to optimize the answers obtained by GPT-4 instead of using other LLMs. According to you, GPT-4 has some inherent flaws that lead to limitations, then to test the optimized results, shouldn't it be better to use other large models, especially after multiple rounds of iterations, instead of just discarding them.
2.2 Regarding the iterative process you mentioned, doesn't it require a human to critique whether this model is good enough for each iteration, or even to screen the entire iteration, which doesn't seem to be a function of automatic screening?
3.1 Why does only provide four domain words to smooth the model? And the 8 patterns from these 4 domains  are used throughout all the experimental design. In my opinion, affect more than just the smoothing of the model?
3.2 Can you introduce the logic of selecting these eight patterns in detail?
3.3 Where does requirement come from, and how does it relate to patterns, "identify specific elements in complex sentences
3.4 Can you introduce in more detail about the content of binary tags, how to annotate, especially for the gold tag binary annotation, I do not see the relevant content of the introduction in Annex C
4.1 This research direction actually has quite a few researchers working on it at the moment. Can you add some more natural datasets and baseline experiments, comparing other methods as well as practical effects?
4.2 Regarding the evaluation metrics section, it is recommended to add evaluation metrics from other genres or articles, such as [1], even though you may consider this evaluation metric to be simple or not as advanced as yours on specific issues, like in [1], i.e., evaluating by distinguishing between labels, is relatively simple, but it would add credibility to the readers.
[1] Wang, Zezhong, et al. "Self-guard: Empower the llm to safeguard itself." arXiv preprint arXiv:2310.15851 (2023).

**Reasons To Accept:**

1. The paper explores a question that deserves to be examined
2. We all know that large models vary in their answers depending on the questions asked, so how they ask questions is critical. The paper proposes an idea of self-challenge, through self-iteration, optimization, and further enable the questions to be asked to the model can be more structured.
3. Although there is room for improvement, the experiments in this paper are relatively complete and can cover most of the questions that need to be verified.

**Reasons To Reject:**

See the above questions

---

> ### Author Rebuttal · Authors · 2024-05-31
>
> Thanks for your valuable review.
>
> **‘Self-Challenge’**
>
> We will revise it by adding *Human-in-the-loop*, referring to the process of LLM finding questions that can challenge themselves with the guidance of humans.
>
>  **Annotation**
>
> For the *unsatisfactory* for each data, we check both label and response:
>
> 1. Incorrect: If any error (in particular factual) is found in GPT4 response, annotators label it as *correct* (or vice versa); annotated responses contain errors, in particular annotators use incorrect GPT4 responses as gold responses without modification
> 2. Unreadable: annotated response is oversimplified particularly if annotators modify GPT4 responses, but annotated responses need GPT4 response as context to understand
> 3. Unnatural: annotators use GPT4 response, but it contains words such as '*As an AI…*'
>
> The binary-label annotation is in Appen C (first para).
>
> **Why GPT4**
>
> We used GPT4 as it is regarded as the most powerful LLM on textual tasks.
>
> Please also see Response to xmpu. We conduct experiments using Llama-3-70B-instruct. But it is less satisfying compared with GPT4.
>
> **Human involvement**
>
> Yes, our work involves the process of human-in-the-loop. In particular, after generating/optimizing error patterns, we evaluate the pattern and obtain feedback from human evaluation (as mentioned in Sec2.2 and 3.2). This process cannot be automated as LLMs do not know what is correct and is incorrect.
>
> **4 domain words**
>
> Those 8 patterns are not summarized from 4 domains, but from 189 seed data, and are then optimized under 4 domain words with human feedback. Here the *smooth* of 4 domains is compared with 1.
>
> We will clarify it as a limitation.
>
> **Pattern selection**
>
> We select 8 *qualified* patterns from the initial 14 ones. As in Sec 2.2, we evaluate the pattern by evaluating the new queries generated by them. The new queries should meet 2 criteria. 1. they should contain important features of the pattern, e.g. an Assumption of Existence (AOE) query should contain non-existent content. 2. the new queries should be challenging enough, e.g. an AOE query should induce LLM to hallucinate.
>
> **Pattern requirement**
>
> The requirement of *identify specific ...* refers to the features in pattern (Fig5, Appen E), which is completely generated by the LLM.
>
> **More LLMs**
>
> Please see Response to xmpu, we benchmark more LLMs.
>
> For all above issues, we will clarify & highlight them in our revision. We would really appreciate it if you could reconsider our paper.

---

> ### Author Response · Authors · 2024-06-05
> **Discussion**
>
> Due to the limited space (2500 characters) for rebuttal, we respond to the main questions. Here, we take the opportunity to engage further discussion.
>
> **Evaluation metrics**
>
> Following [1], we modify our evaluation prompt, asking LLMs to explain why the model-generated response is correct or not [1], and then give a label for the response. In particular, the evaluation prompt is:
>
> ```
> Please first read the below question, its correct answer and a model-generated answer:
>
> Question: {question}
>
> Correct Answer: {answer}
>
> Model-generated Answer: {model_response}
>
> Please explain whether and why the model-generated answer is correct or incorrect and add a tag \"correct\" or \"incorrect\" at the end of your response."
> ```
>
> Similarily, we show the precision, recall, and f1 scores of this metric by comparing its outputs with the human-annotated labels.
>
> On full 1835 instances. Precision 94.57, Recall 78.06, F-1 85.52
> On 115 instances. Precision 90.57, Recall 73.85, F-1 81.36
>
> We see that compared with direct prompting, asking LLMs to first generate an explanation and then a label gives higher precision and lower recall and F-1 scores. We assume that this can be because the LLMs can generate inappropriate explanations for those questions (in particular those questions are challenging to LLMs),  and subsequently give a wrong label.
>
> For example, given the question:
>
> ```
> Suppose there is a rule that patients with Long QT syndrome can only participate in sports with a prime-numbered amount of players on the field. If a patient wants to join a sports team where the number of players is doubled each season, which sport should they choose to maximize their chances of participation in the first three seasons?
> ```
>
> And the GPT-4 answer is:
>
> ```
> To maximize the chances of participation in the first three seasons, we need to find a sport where the number of players on the field is a prime number in each of the first three seasons. Let's denote the initial number of players as x. Then, the number of players in the first three seasons will be x, 2x, and 4x.
>
> We need to find a value of x such that x, 2x, and 4x are all prime numbers. Let's analyze the possibilities:
>
> 1. If x is an even number, then 2x and 4x will also be even numbers. The only even prime number is 2, but if x = 2, then 2x = 4, which is not a prime number. So, x cannot be an even number.
>
> 2. If x is an odd number, then 2x will be an even number. As mentioned earlier, the only even prime number is 2. However, if 2x = 2, then x = 1, which is not a prime number.
>
> Based on this analysis, there is no sport that meets the criteria of having a prime-numbered amount of players on the field for the first three seasons when the number of players is doubled each season.
> ```
> which is correct as labeled by both human and our evaluation metric.
>
> However, the above prompting evaluation gives an explanation:
>
> ```
> ​​The model-generated answer is incorrect. The explanation provided by the model is accurate in stating that if x is an even number, then 2x and 4x will also be even and therefore cannot be prime (except for the number 2). However, the model then incorrectly states that if x is an odd number, then 2x will be an even number and cannot be prime, which is true, but it incorrectly concludes that there is no sport that meets the criteria.
>
> The fundamental error in the model-generated answer is the assumption that there could be a prime number x that, when doubled or quadrupled, results in other prime numbers. This is impossible because, except for the number 2, all prime numbers are odd, and multiplying an odd prime number by 2 or 4 will always result in an even number greater than 2, which cannot be prime.
>
> The correct answer clearly states that it is impossible for x, 2x, and 4x to all be prime numbers because 2x and 4x will always be composite if x is greater than 2. Therefore, there is no sport that meets the criteria of having a prime-numbered amount of players on the field for the first three seasons when the number of players is doubled each season.
>
> Incorrect
> ```
> where the explanation incorrectly reasons ```the model then incorrectly states that if x is an odd number ... will always result in an even number greater than 2, which cannot be prime.```
>
> We will include the above case and result as a supplement material in our revision.
>
> Wang, Zezhong, et al. "Self-guard: Empower the llm to safeguard itself." arXiv preprint arXiv:2310.15851 (2023).

---

### Decision · Program_Chairs · 2024-07-10

**Decision:**

Accept

**Comment:**

The reviewers found the paper clear and supported its motivation (better identification and measurement of LLM shortcomings). Reviewers liked the idea of including humans in the loop in the process of finding model challenges. The experiments in the paper seem to be clearly explained and executed, but some reviewers point out limitations of the Self-Chal benchmark (especially the types of tasks/problems it covers and its reliance on GPT-4).

During the discussion period, the authors provided results from some follow-up experiments to address some of these issues (e.g., benchmarking more models, getting human performance numbers), which should be added to the next version of the paper. However, some of the limitations are unavoidable, given the design of the framework (e.g., the type of tasks it can include).